# Watch Before You Answer: Learning from Visually Grounded Post-Training

## Abstract

It is critical for vision-language models (VLMs) to comprehensively understand visual, temporal, and textual cues. However, despite rapid progress in multimodal modeling, video understanding performance still lags behind text-based reasoning. In this work, we find that progress is even worse than previously assumed: commonly reported long video understanding benchmarks contain 40-60% of questions that can be answered using text cues alone. Furthermore, we find that these issues are also pervasive in widely used post-training datasets, potentially undercutting the ability of post-training to improve VLM video understanding performance. Guided by this observation, we introduce VIDGROUND as a simple yet effective solution: using only the actual visually grounded questions without any linguistic biases for post-training. When used in tandem with RL-based post-training algorithms, this simple technique improves performance by up to 6.2 points relative to using the full dataset, while using only 69.1% of the original post-training data. Moreover, we show that data curation with a simple post-training algorithm outperforms several more complex post-training techniques, highlighting that data quality is a major bottleneck for improving video understanding in VLMs. These results underscore the importance of curating post-training data and evaluation benchmarks that truly require visual grounding to advance the development of more capable VLMs.

## 1 Introduction

Video understanding is vital for real-world AI, with applications including autonomous driving, online tutorial development, assistive robotics, and movie analysis, where models must accurately integrate visual, temporal, and textual cues (Tang et al., 2025; Bahl et al., 2023; Elhenawy et al., 2025). Despite recent advances in vision-language models (VLMs), driven by larger video training datasets (Achiam et al., 2023; Comanici et al., 2025) and improved multimodal alignment techniques (Lin et al., 2024b; Xue et al., 2023; Wang et al., 2025a), performance has lagged behind text-based reasoning, especially for tasks involving long-context video understanding such as MMVU (Zhao et al., 2025) and VideoMME (Fu et al., 2025a).

Here we show that the community's progress in improving video understanding in VLMs is even worse than initially thought, with a majority of the gains coming from models' abilities to answer questions *without access to the video* (Fig. 1).

This phenomenon, known as "linguistic shortcutting," has been well established in Visual Question Answering (VQA) as a serious problem. As a result, video understanding benchmark designers have tried to avoid these pitfalls in benchmarks such as VideoMME (Fu et al., 2025a) by filtering for questions that could be answered by the leading foundation model at the time without the video. However, as VLMs become stronger, we find that their gains come from being able to answer a larger portion of the benchmark without access to the video (Fig. 1), with their ability to answer video-based questions sometimes *worsening*. Consequently, these benchmarks are now problematic for measuring improvements in genuine video understanding.

We find that this problem is pervasive not just for evaluation benchmarks, but also for the most commonly used video understanding post-training datasets. Guided by this observation, we introduce VIDGROUND, a simple yet effective approach to post-training VLMs: using only visually grounded questions. Although

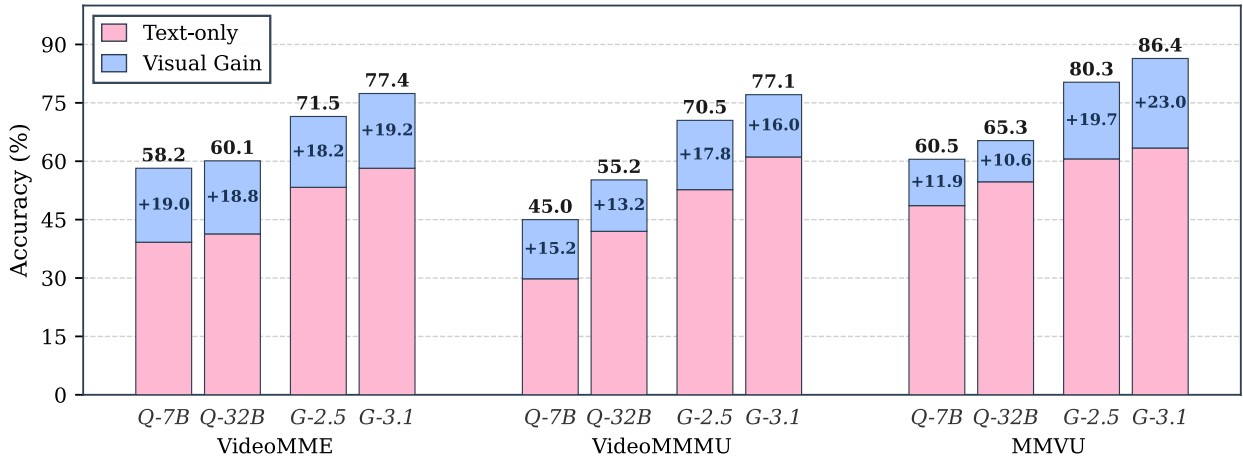

Figure 1: Performance decomposition on three video understanding benchmarks for four frontier VLMs: Qwen2.5-VL-7B and 32B (Q-7B, Q-32B) (Bai et al., 2025), and Gemini-2.5-Pro and 3.1-Pro (G-2.5, G-3.1) (Comanici et al., 2025; Google, 2026). Pink bars show text-only accuracy (no video input); blue bars show the additional visual gain from video access. The majority of benchmark performance comes from language priors rather than visual comprehension. Moreover, scaling up model size or version improves text-only reasoning but visual gain often remains flat or even *decreases*.

this strategy uses only 69.1% of the post-training data, it leads to improvements of up to 6.2 points in video understanding performance relative to post-training on the full dataset. More surprisingly, this simple approach also outperforms several more advanced RL-based post-training strategies, including methods that employ token-level importance weighting (Dang et al., 2025), long-video sequence scaling (Chen et al., 2025c), and adaptive test-time frame selection (Wang et al., 2025b). Overall, this suggests that a major bottleneck for improving VLM video understanding via post-training rests in the data. The effectiveness of using visually grounded questions suggests a deep well of improvement with algorithmic solutions that maximize this grounding signal.

Our contributions are as follows:

- We systematically analyze linguistic biases in video understanding benchmarks and post-training datasets, finding that 40–60% of questions in popular benchmarks can be answered from text alone across multiple frontier models.

- We introduce VIDGROUND, a simple data curation approach for post-training that selects only visually grounded questions—those that genuinely require visual understanding to answer.

- We show that post-training on only visually grounded data with a simple RL algorithm outperforms several more complex post-training techniques, demonstrating that data quality is a major bottleneck for improving video understanding in VLMs.

## 2 Related work

### 2.1 Language priors in VLMs

Linguistic shortcutting for Visual Question Answering (VQA) has been known to be an issue since the seminal work of Goyal et al. (2017), which demonstrated that early VQA models learned to rely more on text than vision for answering questions. Since then, many recent studies (Rahmanzadehgervi et al., 2024; Campbell et al., 2024; Huang et al., 2025; Xu et al., 2025; Chandhok et al., 2025; Zhang et al., 2024a; Tong et al., 2024; Peng et al., 2024; Ranasinghe et al., 2024) have shown that modern VLMs still exhibit clear weaknesses on basic vision-centric tasks, such as spatial reasoning, object counting, geometric perception, visual analogy,

and fine-grained recognition. Although their visual encoders are powerful, VLMs significantly underperform their visual encoders on tasks like image classification (Zhang et al., 2024c) and depth estimation (Fu et al., 2025b). Other analyses (Luo et al., 2024; Parashar et al., 2024; Miller et al., 2024; Alhamoud et al., 2025; Lee et al., 2025; Vo et al., 2025) report that VLMs exhibit a significant reliance on language priors rather than true visual grounding. As shown by Bleeker et al. (2024), VLMs can learn shortcuts in which they rely on easily-discriminative but non-task-optimal features instead of capturing all the shared vision-language information they should.

However, despite many studies investigating this phenomenon for VQA, relatively little work has investigated it for video understanding. Given that video understanding should require synthesizing visual information across multiple frames, it may be less likely for linguistic shortcutting to present a major problem. For example, Park et al. (2025) and Wu et al. (2025) found that VLMs exhibit modality bias in favor of linguistic input in videos when subtitles are available, but did not investigate linguistic bias in the absence of subtitles (i.e. when given just the question text). Here we investigate linguistic shortcutting when neither subtitles nor the video is available.

## 2.2 Strategies to improve VLM performance

Early attempts to mitigate linguistic shortcutting were applied to VQA models. These included augmenting how data is used for training by changing its weighting based on how easy it is to answer via text alone (Niu et al., 2021; Cadene et al., 2019) or by changing the training objective to prioritize visual information (Ramakrishnan et al., 2018; Liang et al., 2021).

Recent work instead focuses on *post-training* VLMs to improve their visual capabilities. Supervised fine-tuning (SFT) and reinforcement learning (RL) are the dominant paradigms for post-training. Chen et al. (2025a) demonstrated that generally RL is superior to SFT for post-training multimodal models, so we focus on the RL family of approaches. In the video domain, Video-R1 (Feng et al., 2025) represents the first systematic exploration of the RL paradigm for video reasoning. Video-R1 introduces a temporal contrastive auxiliary reward to Group Relative Policy Optimization (GRPO) (Shao et al., 2024) which has shown great success for text. Video-R1 further integrates curated video data and image-based reasoning samples, constructing Video-R1-CoT-165k for supervised warm-up and Video-R1-260K for reinforcement learning. Other RL-style approaches include LongVILA-R1 (Chen et al., 2025c) which scales the R1-style GRPO framework to genuinely long-video settings, TW-GRPO (Dang et al., 2025) which computes token-level importance weights and down-weights redundant ones, and Video-RTS (Wang et al., 2025b) which introduces a sparse-to-dense test-time scaling strategy for improved efficiency during RL-based post-training.

Here, we demonstrate how using visually grounded data in combination with RL-based post-training can outperform these approaches for improving VLM video understanding.

## 3 Analyzing linguistic biases in video understanding datasets

It is well known that linguistic biases are pervasive in VQA benchmarks. What about video understanding? Video understanding benchmarks such as VideoMME (Fu et al., 2025a) were explicitly designed to avoid linguistic shortcutting, but were they successful?

We first investigate the prevalence of linguistic biases in video understanding benchmarks and post-training datasets, showing that substantial portions can be answered without video input. To analyze the quality of existing video understanding benchmarks and post-training datasets, we conduct a simple yet effective experiment: evaluating VLM performance on video datasets by providing only questions and answer choices while withholding all visual content. We denote questions that can be answered correctly without accessing any visual content as text-only answerable (TA) questions, and the remainder as visually grounded (VG) questions.

As shown in Table 1, we find that a substantial proportion of questions across popular video understanding benchmarks can be answered correctly by frontier models using text alone. For instance, VideoMME and MMVU (multiple-choice) contain 48.2% and 57.1% TA questions respectively as measured with GPT-

Table 1: Text-only Answerability (TA) across video understanding benchmarks for frontier models. Each model receives only the question text and answer options—no video input—yet achieves accuracy far above random chance. $(+x)$ denotes improvement relative to random choice. Results indicate that 40–60% of benchmark questions can be answered from text alone, revealing substantial linguistic bias in existing video understanding benchmarks.

| Model | VideoMME | VideoMMMU | MMVU |
|---|---|---|---|
| Random Choice | 25.0 | 9.8 | 19.8 |
| GPT-4o (Achiam et al., 2023) | 47.0 (+22.0) | 38.6 (+28.8) | 46.6 (+26.8) |
| GPT-5-mini (OpenAI, 2025) | 45.2 (+20.2) | 37.9 (+28.1) | 53.3 (+33.5) |
| GPT-5 (OpenAI, 2025) | 48.2 (+23.2) | 41.0 (+31.2) | 57.1 (+37.3) |
| Gemini-2.5-Pro (Comanici et al., 2025) | 53.3 (+28.3) | 52.7 (+42.9) | 60.6 (+40.8) |
| Gemini-3.1-Pro (Google, 2026) | 58.2 (+33.2) | 61.1 (+51.3) | 63.4 (+43.6) |
| Claude-Sonnet-4.5 (Anthropic, 2025) | 47.7 (+22.7) | 44.3 (+34.5) | 55.4 (+35.6) |
| Claude-Opus-4.6 (Anthropic, 2026) | 51.3 (+26.3) | 52.7 (+42.9) | 61.0 (+41.2) |

5 (OpenAI, 2025) or 58.2% and 63.4% as measured with Gemini-3.1-Pro (Google, 2026). These numbers are substantially higher than chance performance and indicate that a large proportion of questions in these benchmarks can be answered correctly without visual information. Figure 2b further illustrates this breakdown, showing the proportion of VG versus TA questions across benchmarks along with the distribution of TA subcategories.

Similar patterns emerge in video understanding post-training datasets: Video-R1-260K (Feng et al., 2025) contains 30.9% TA questions (as measured with GPT-5-mini), suggesting that nearly one-third of the post-training data may not require genuine visual understanding.

These findings reveal significant biases in both current video understanding benchmarks and post-training datasets across multiple frontier models, with critical implications for model development and evaluation. When a substantial proportion of TA questions exists in evaluation benchmarks, model performance becomes artificially inflated, causing benchmark scores to misrepresent true video understanding capabilities (see Fig. 1). More critically, when video understanding post-training datasets contain high proportions of TA questions, they inevitably exacerbate linguistic biases in VLMs, leading models to develop stronger language priors rather than improved visual grounding.

### 3.1 Analysis of text-only answerable questions

Moreover, we identify the four most common types of linguistic biases and discuss how they can encourage linguistic shortcuts in video understanding tasks.

Within common post-training data such as Video-R1-260K (Feng et al., 2025) and standard, widely used video understanding benchmarks (Fu et al., 2025a; Hu et al., 2025), we identified four common categories of TA questions (illustrated in Fig. 2a) that are answerable by VLMs without visual input.

**Textual shortcuts and linguistic cues.** Questions contain surface-level hints that reveal the answer without visual grounding. For instance, when asked "How does the cookie change after being decorated?", the option "It becomes more colorful" can be inferred linguistically, as the word "decorated" naturally implies adding visual elements or colors.

**External knowledge.** Questions can be answered using commonsense or world knowledge alone. For example, "What does the person rely on for support while descending the cliffside?" can be correctly answered as "A rope" based on common knowledge about rappelling and climbing activities, without observing the video content.

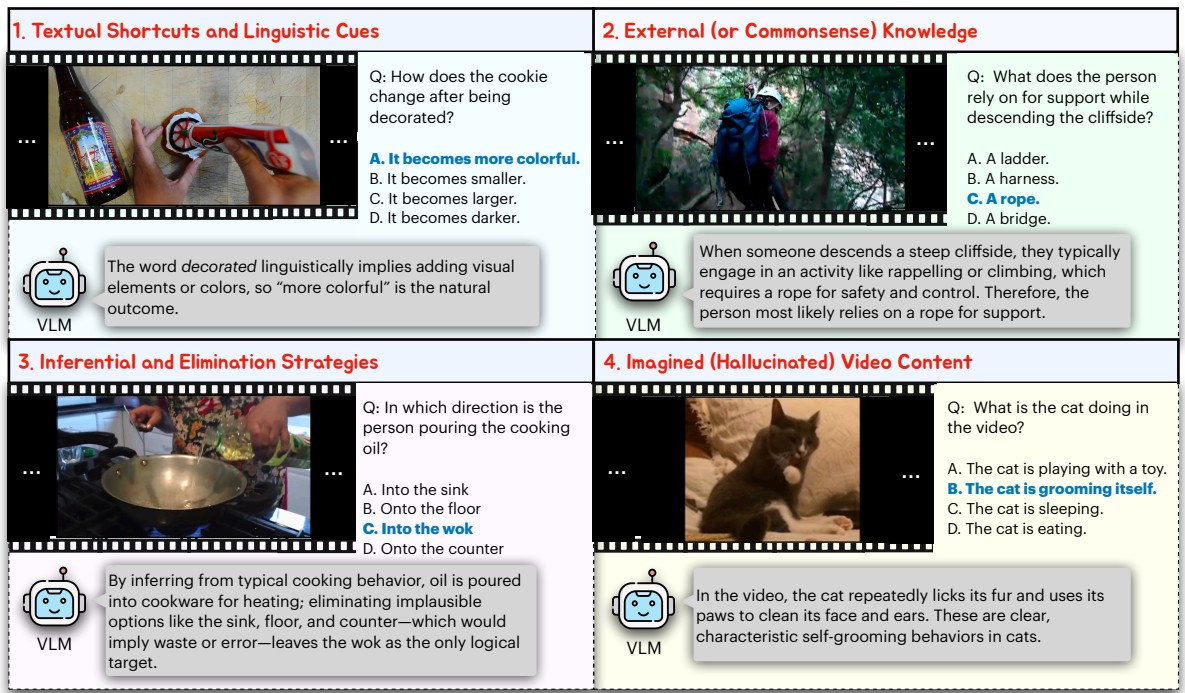

(a) Common categories of TA questions that allow VLMs to answer correctly without visual grounding. Examples are drawn from Video-R1-260K (Feng et al., 2025), with responses from GPT-5-mini.

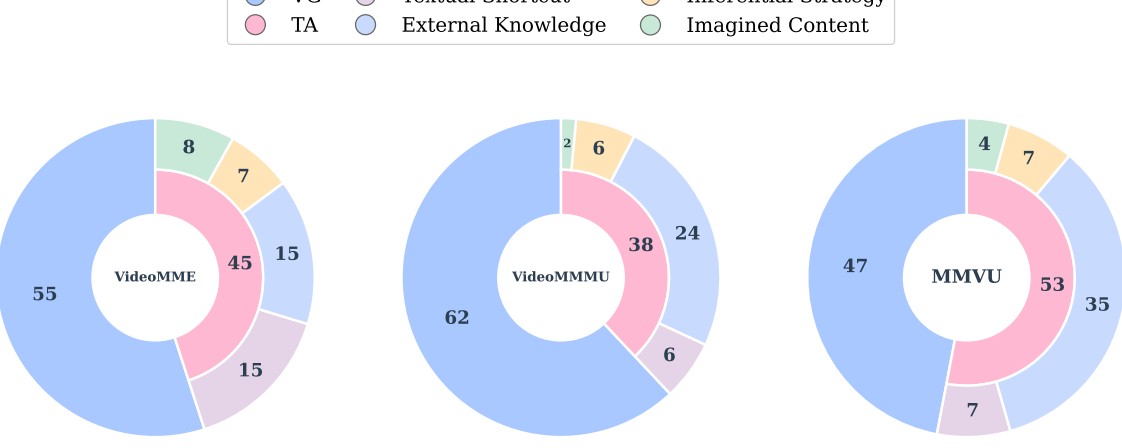

(b) Breakdown of TA and visually grounded (VG) items for VideoMME, VideoMMMU, and MMVU, classified using GPT-5-mini. TA items are further categorized into four reasoning types (see §3.1). Numbers indicate the percentage of examples in each category.

Figure 2: Analysis of text-only answerable (TA) questions in video understanding benchmarks and post-training data. (a) We identify four common categories of linguistic shortcuts—textual cues, external knowledge, inferential strategies, and imagined content—that allow VLMs to answer correctly without watching the video. (b) These TA questions comprise 38–53% of popular benchmarks (classified using GPT-5-mini), with external knowledge being the dominant category in VideoMMMU and MMVU.

**Inferential and elimination strategies.** Questions allow models to succeed through logical reasoning and elimination of implausible options. In the question "In which direction is the person pouring the cooking

oil?", options like "into the sink," "onto the floor," and "onto the counter" can be eliminated as they imply waste or error, leaving "into the wok" as the only logical choice.

**Imagined (hallucinated) video content.** Models generate plausible video scenarios based solely on questions and options, which happen to align with actual content. For instance, when asked "What is the cat doing in the video?", a model might correctly guess "The cat is grooming itself" by imagining typical cat behaviors, even without visual evidence.

These categories of TA questions reveal fundamental issues for both evaluation and post-training.

For evaluation, as frontier models become more powerful, their ability to take advantage of external knowledge and to use inferential and elimination reasoning strategies will only increase. This will further inflate model performance without reflecting improvements in genuine video understanding.

For post-training, when VLMs are post-trained on data containing substantial proportions of such TA questions, they may learn to exploit textual patterns and world knowledge instead of establishing robust vision-language associations, ultimately undermining their video understanding abilities.

These observations motivate a straightforward hypothesis: post-training on visually grounded data— questions that genuinely require visual understanding—should yield better video understanding than training on data contaminated by linguistic shortcuts. In the next section, we describe our approach to curating high-quality visually grounded post-training data, and in Section 5 we empirically validate that visually grounded training data leads to stronger video understanding performance.

## 4 VidGround: a simple approach to post-training

Guided by these analyses, we introduce VIDGROUND, a simple technique for improving video understanding in VLMs through post-training. VIDGROUND combines reinforcement learning techniques for post-training (described in §4.1) with a simple data curation method (described in §4.2). While VIDGROUND can be applied to any base VLM, we adopt Qwen2.5-VL-7B-Instruct (Bai et al., 2025) for its video understanding capabilities and computational efficiency.

### 4.1 RL for video understanding post-training

We use reinforcement learning (RL) for post-training based on recent evidence that RL improves underlying visual recognition capabilities (Chen et al., 2025b) while exhibiting less catastrophic forgetting than supervised fine-tuning (SFT) (Chu et al., 2025).

**Optimization objective.** We adopt Group Relative Policy Optimization (GRPO) (Shao et al., 2024) augmented with techniques from DAPO (Yu et al., 2025) and temporal-aware rewards from Video-R1 (Feng et al., 2025). Specifically, we employ token-level policy gradient loss with asymmetric clipping (increasing the value of $\varepsilon_{\mathrm{h}}$) to make the training more efficient and stable. Our objective is formulated as:

$$\mathcal{J}(\theta) = \mathbb{E}_{(q,a)\sim\mathcal{D},\{o_i\}_{i=1}^{G}\sim\pi_{\theta_{\mathrm{old}}}}(\cdot \mid q)$$

$$\left[\frac{1}{\sum_{i=1}^{G}|o_i|}\sum_{i=1}^{G}\sum_{t=1}^{|o_i|}\ell_i^{\mathrm{clip}}(\theta) - \beta\mathbb{D}_{\mathrm{KL}}\left(\pi_\theta\|\pi_{\mathrm{ref}}\right)\right] \quad (1)$$

where

$$\ell_i^{\mathrm{clip}}(\theta) = \min\left(\rho_i(\theta)\,\hat{A}_i,\ \mathrm{clip}\left(\rho_i(\theta), 1-\varepsilon_{\mathrm{l}}, 1+\varepsilon_{\mathrm{h}}\right)\hat{A}_i\right),$$

$$\rho_i(\theta) = \frac{\pi_\theta(o_i \mid q)}{\pi_{\theta_{\mathrm{old}}}(o_i \mid q)}, \quad \hat{A}_i = \frac{r_i - \mathrm{mean}(\mathbf{r})}{\mathrm{std}(\mathbf{r})}.$$

Here, $q$ denotes the video-question input, $o_i$ represents the $i$-th sampled response from a group of $G$ samples, $\hat{A}_i$ is the advantage computed from reward $r_i$, and $\beta$ controls the KL term relative to the reference policy $\pi_{\text{ref}}$. $\varepsilon_l$ and $\varepsilon_h$ are the lower and upper clipping bounds, respectively.

### 4.2 Post-training data curation

We curate our post-training data from Video-R1-260K (Feng et al., 2025), which comprises 116,248 Video QA and 146,823 Image QA instances spanning diverse video and image understanding scenarios. Our goal is to select only visually grounded (VG) questions—those that genuinely require visual understanding to answer.

**Selection pipeline.** To identify VG questions, we prompt GPT-5-mini (OpenAI, 2025) with only the question text and answer options (no visual input) and retain only questions it cannot answer correctly. This text-only evaluation step selects 181,710 visually grounded samples (69.1% of the original dataset)— questions that require genuine visual understanding. We note that this selection is not an artifact of a single model: of the 181,710 VG questions selected by GPT-5-mini, 85% are also unanswerable by Qwen2.5-VL-7B (Bai et al., 2025) in text-only mode, confirming that the retained questions genuinely require visual input. Furthermore, applying circular evaluation with Gemini-3.1-Pro (Google, 2026)—rotating MCQ answer option positions—yields 97% agreement across permutations, confirming that the selection is robust to positional bias.

**Training data variants.** To investigate the causal effect of linguistic biases on post-training, we compare two data variants. The **Full** variant (263,071 samples) represents standard post-training without curation. The **VG** variant (181,710 samples) consists solely of visually grounded questions that GPT-5-mini cannot answer from text alone. VIDGROUND is the combination of the **VG** dataset with the RL-based post-training outlined above.

## 5 Experiments

We first describe our experimental setup (§5.1), then compare VIDGROUND against strong baselines (§5.2). We further analyze the contribution of each post-training dataset variant (§5.3) and conclude with a qualitative comparison of reasoning chains between our model and the baselines (§5.4).

### 5.1 Experimental setup

**Post-training configuration** We uniformly sample 16 frames per video and post-train Qwen2.5-VL-7B for 700 steps using the GRPO objective described in §4.1. Our primary results (Table 2) are obtained using VIDGROUND by post-training on the VG variant, which contains only visually grounded questions. To investigate the impact of post-training data composition, we report the performance of models post-trained on the Full variant in Table 3.

**Benchmarks** We evaluate on three established video understanding benchmarks: VideoMME (Fu et al., 2025a), a comprehensive, general-purpose benchmark spanning perception and reasoning; VideoMMMU (Hu et al., 2025), focused on expert-level, multi-disciplinary video reasoning; and MMVU (Zhao et al., 2025), emphasizing college-level, knowledge-intensive video comprehension. For MMVU, we evaluate on multiple-choice questions to ensure consistency and fair comparison. Following standard protocols, we report accuracy scores for all benchmarks.

**Baseline post-training approaches** We compare our approach against other strong 7B-scale post-training techniques including LongVILA-R1 (Chen et al., 2025c), TW-GRPO (Dang et al., 2025), Video-RTS (Wang et al., 2025b), and Video-R1 (Feng et al., 2025). We also compare to our base model, Qwen2.5-VL-7B (Bai et al., 2025), and its SFT variant (Qwen2.5-VL-7B-SFT). Notably, except for LongVILA-R1, all baseline models are originally post-trained from Qwen2.5-VL-7B using publicly available post-training data.

Table 2: Performance comparison of 7B-scale post-training methods on three video understanding benchmarks (VideoMME, VideoMMMU, MMVU). All methods except LongVILA-R1 are post-trained from Qwen2.5-VL-7B. Models are evaluated using 16, 32, and 64 frames per video. VIDGROUND is post-trained with GRPO on visually grounded (VG) data only. Avg. columns report mean accuracy on the full benchmarks (Full) and on VG question subsets—questions that require video to answer (VG). Deltas show improvement over Qwen2.5-VL-7B: $(+x)$ and $(-x)$. Bold indicates best; highlighted rows are ours.

| Frames | Method | VideoMME | VideoMMMU | MMVU | Avg. | |
|---|---|---|---|---|---|---|
| | | | | | Full | VG |
| 16 | Qwen2.5-VL-7B | 58.2 | 45.0 | 60.5 | 54.6 | 42.9 |
| | TW-GRPO | 58.2 | **48.6** | 61.8 | 56.2 (+1.6) | 44.1 (+1.2) |
| | LongVILA-R1-7B | 55.5 | 38.8 | 59.1 | 51.1 (−3.5) | 39.7 (−3.2) |
| | Video-RTS | **58.7** | 47.1 | 61.8 | 55.9 (+1.3) | 43.5 (+0.6) |
| | Qwen2.5-VL-7B-SFT | 58.2 | 43.1 | 51.3 | 50.9 (−3.7) | 41.1 (−1.8) |
| | *Video-R1* | 56.9 | 44.7 | 54.5 | 52.0 (−2.6) | 41.7 (−1.2) |
| | **VidGround** | **58.7** | 47.4 | **64.2** | 56.8 (+2.2) | 45.2 (+2.3) |
| 32 | Qwen2.5-VL-7B | 60.7 | 45.4 | 62.3 | 56.1 | 44.4 |
| | TW-GRPO | 61.2 | 47.9 | 63.1 | 57.4 (+1.3) | 45.9 (+1.5) |
| | LongVILA-R1-7B | 60.2 | 40.7 | 61.5 | 54.1 (−2.0) | 42.9 (−1.5) |
| | Video-RTS | 61.3 | 47.7 | 65.0 | 58.0 (+1.9) | 46.3 (+1.9) |
| | Qwen2.5-VL-7B-SFT | 60.7 | 47.8 | 51.0 | 53.2 (−2.9) | 44.6 (+0.2) |
| | *Video-R1* | 60.2 | 45.4 | 56.2 | 53.9 (−2.2) | 43.1 (−1.3) |
| | **VidGround** | **61.5** | **48.3** | **65.8** | 58.4 (+2.4) | 47.6 (+3.2) |
| 64 | Qwen2.5-VL-7B | 62.3 | 46.6 | 62.6 | 57.2 | 46.3 |
| | TW-GRPO | 62.7 | 48.3 | 64.2 | 58.4 (+1.2) | **48.2** (+1.9) |
| | LongVILA-R1-7B | 61.6 | 41.2 | 58.8 | 53.9 (−3.3) | 42.0 (−4.3) |
| | Video-RTS | 62.9 | 46.4 | 63.9 | 57.7 (+0.5) | 46.4 (+0.1) |
| | Qwen2.5-VL-7B-SFT | 62.2 | 47.6 | 55.4 | 55.1 (−2.1) | 45.8 (−0.5) |
| | *Video-R1* | 61.2 | 45.4 | 53.2 | 53.3 (−3.9) | 42.9 (−3.4) |
| | **VidGround** | **63.4** | **49.4** | **65.6** | 59.4 (+2.3) | 47.4 (+1.6) |

## 5.2 Results

**Across post-training approaches** Table 2 presents our main results compared to strong 7B-scale post-training methods for video understanding at 16, 32, and 64 frames. Compared to Video-R1 (Feng et al., 2025), which trains on the full unfiltered dataset, VIDGROUND improves by an average of 4.8, 4.6, and 6.2 points on Full Avg at 16, 32, and 64 frames respectively, while using only 69.1% of the training data. On visually grounded (VG) questions—those requiring video to answer—the gains are 3.5, 4.5, and 5.0 points on VG Avg. Relative to the base model Qwen2.5-VL-7B, VIDGROUND improves by 2.2, 2.4, and 2.3 points on Full Avg. At all frame settings, VIDGROUND maintains the highest Full Avg performance among all baselines. These results demonstrate our simple data curation technique can effectively improve video understanding capability of models.

**Across datasets** The benefits of VIDGROUND are particularly pronounced on benchmarks that emphasize visual comprehension. On MMVU, which requires fine-grained visual understanding across diverse domains, VIDGROUND outperforms Qwen2.5-VL-7B by 3.0 points at 64 frames. Similarly, on VideoMME, which includes many perception-intensive tasks, VIDGROUND achieves the highest performance across all frame settings. On average, VIDGROUND provides consistent gains across all datasets and temporal resolutions. These results indicate that visually grounded post-training data benefits models across diverse benchmarks. Importantly, visually grounded post-training does not degrade image understanding capabilities: VIDGROUND improves over Qwen2.5-VL-7B on MME (648.9 vs. 624.3) and MMMU (58.7 vs. 56.7),

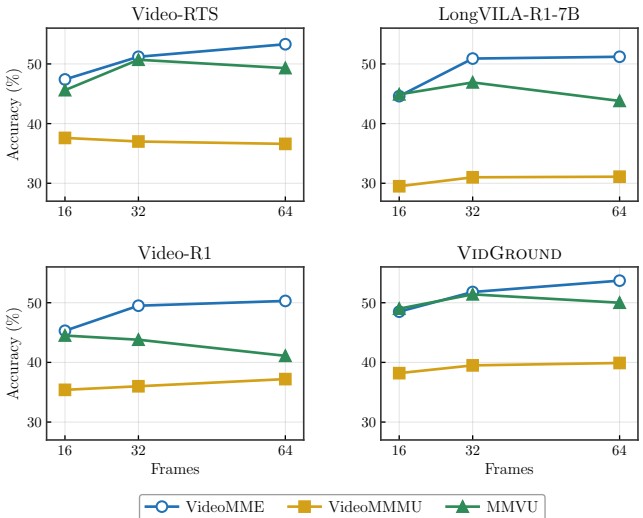

Figure 3: Accuracy on visually grounded (VG) questions—questions that cannot be answered from text alone—across 16, 32, and 64 frames on VideoMME, VideoMMMU, and MMVU for four post-training methods. VG questions are identified by retaining only questions that cannot be answered from text alone, as classified by GPT-5-mini (see §3). VIDGROUND shows the strongest overall frame-scaling behavior, while baselines such as LongVILA-R1-7B and Video-R1 plateau or degrade on MMVU, and Video-RTS drops on VideoMMMU. This suggests that models post-trained on data containing linguistic shortcuts do not effectively leverage additional visual information.

indicating that curating post-training data for visual grounding does not narrow the training distribution in ways that harm non-video tasks.

**Across frames** We also investigate the models' behaviors as the number of frames increases (Fig. 3). While VIDGROUND generally improves from 16 to 64 frames, several baselines' performance drops with additional frames. Specifically, on MMVU, LongVILA-R1-7B decreases by 3.1 points from 32 to 64 frames, Video-R1 by 2.7 points over the same range. On VideoMMMU, Video-RTS drops by 0.4 points from 32 to 64 frames. These performance drops with increasing numbers of frames, despite access to more visual information, suggest that many existing post-trained models do not effectively leverage additional visual information. We hypothesize that this stems from the substantial proportion of text-only answerable questions in their post-training data, which encourages reliance on linguistic information. In contrast, VIDGROUND shows the most consistent improvements with more frames, implying that post-training on visually grounded data enables the model to leverage temporal and visual cues more effectively.

### 5.3 Ablation study

Table 3 presents our ablation study comparing post-training data variants. We compare three configurations: GRPO trained on Full data, i.e., Video-R1 (Feng et al., 2025), and GRPO on VG-only data with and without asymmetric clipping (+clip-higher). Overall, we find little impact of using asymmetric clipping, but large impacts depending on the data used for post-training.

**Less is more** VIDGROUND (GRPO+VG in Table 3), post-trained on 181K visually grounded samples, consistently outperforms the model trained on the full 263K dataset across all frame settings, using only 69.1% of the post-training data. Compared to GRPO trained on the full dataset, VIDGROUND achieves average improvements of 4.8, 4.6, and 6.2 points on Full Avg, and 3.5, 4.5, and 5.0 points on VG Avg, at 16, 32, and 64 frames, respectively. These results suggest that curating a post-training set focused on visually grounded reasoning allows models to learn from fewer but more informative samples, improving both performance on video-grounded tasks and overall training efficiency.

Table 3: Ablation study on post-training data composition. We compare GRPO trained on two data variants of Video-R1-260K (Feng et al., 2025): Full (all 263K samples) and VG (181K visually grounded samples—questions that require visual understanding to answer). +clip-higher denotes asymmetric clipping (see §4.1). Avg. columns report mean accuracy on the full benchmarks (Full) and on VG question subsets—questions that require video to answer (VG). Deltas show improvement over Qwen2.5-VL-7B: (+x) and (−x). Training on VG data consistently outperforms the Full variant despite using 31% less data. Highlighted rows are ours.

| Frames | Method | Data | VideoMME | VideoMMMU | MMVU | Avg. | |
|--------|--------|------|----------|-----------|------|------|------|
| | | | | | | Full | VG |
| 16 | Base | - | 58.2 | 45.0 | 60.5 | 54.6 | 42.9 |
| | GRPO | Full | 56.9 | 44.7 | 54.5 | 52.0 (−2.6) | 41.7 (−1.2) |
| | GRPO | VG | **58.7** | 47.4 | **64.2** | 56.8 (+2.2) | 45.2 (+2.3) |
| | +clip-higher | VG | 58.2 | **47.7** | 63.6 | 56.5 (+1.9) | 45.1 (+2.2) |
| 32 | Base | - | 60.7 | 45.4 | 62.3 | 56.1 | 44.4 |
| | GRPO | Full | 60.2 | 45.4 | 56.2 | 53.9 (−2.2) | 43.1 (−1.3) |
| | GRPO | VG | **61.5** | 48.3 | **65.8** | 58.5 (+2.4) | 47.6 (+3.2) |
| | +clip-higher | VG | 61.4 | **49.2** | 64.2 | 58.3 (+2.2) | 46.8 (+2.4) |
| 64 | Base | - | 62.3 | 46.6 | 62.6 | 57.2 | 46.3 |
| | GRPO | Full | 61.2 | 45.4 | 53.2 | 53.3 (−3.9) | 42.9 (−3.4) |
| | GRPO | VG | 63.4 | **49.4** | **65.6** | 59.5 (+2.3) | 47.9 (+1.6) |
| | +clip-higher | VG | **63.5** | 48.6 | 65.3 | 59.1 (+1.9) | 48.5 (+2.2) |

**Visually grounded training enables consistent frame-scaling** Models trained on VG data show steady improvement as the number of frames increases (e.g., 56.8 to 58.5 to 59.5 on Full Avg for GRPO with VG), whereas the model trained on the full dataset exhibits inconsistent scaling and minimal gains (e.g., 52.0 to 53.9 to 53.3 for GRPO with Full). This pattern is even more pronounced on VG evaluation: GRPO with VG improves from 45.2 to 47.6 to 47.9 on VG Avg, while GRPO with Full stalls at 41.7 to 43.1 to 42.9, declining from 32 to 64 frames. This contrast highlights that visually grounded post-training allows models to more effectively leverage temporal information as additional frames are provided, while linguistic bias leads to plateauing or diminishing returns even when more visual data is available.

## 5.4 Qualitative analysis

To further investigate the benefits of VIDGROUND, we analyzed the reasoning patterns of Video-R1 (Feng et al., 2025) and VIDGROUND on multiple video-dependent samples from VideoMMMU. Figure 4 shows a representative example illustrating the differences in how the two models process visual and textual information. We observe that Video-R1 relies heavily on textual context, producing answers based on abstract reasoning about art concepts without referencing the video. In contrast, VIDGROUND grounds its analysis in actual video content (e.g., identifying specific visual elements such as lines, shapes, and colors), leading to a correct answer. This pattern is consistent across diverse expert domains—including medical imaging, structural engineering, chemistry, and public health (additional examples in the supplementary material). Across all analyzed instances, VIDGROUND systematically anchors its reasoning in observed video content, whereas Video-R1 defaults to analyzing questions through prior knowledge and linguistic cues. Notably, even when both models arrive at the correct answer, their reasoning processes differ fundamentally—VIDGROUND derives the answer from video content while Video-R1 reaches it through text-based elimination—indicating that accuracy metrics alone cannot fully capture whether a model genuinely leverages visual information.

## 6 Discussion

Notably, when evaluated with frontier models, some of the most popular video understanding benchmarks, such as VideoMME (Fu et al., 2025a) and VideoMMMU (Hu et al., 2025), contain 40–60% of questions that

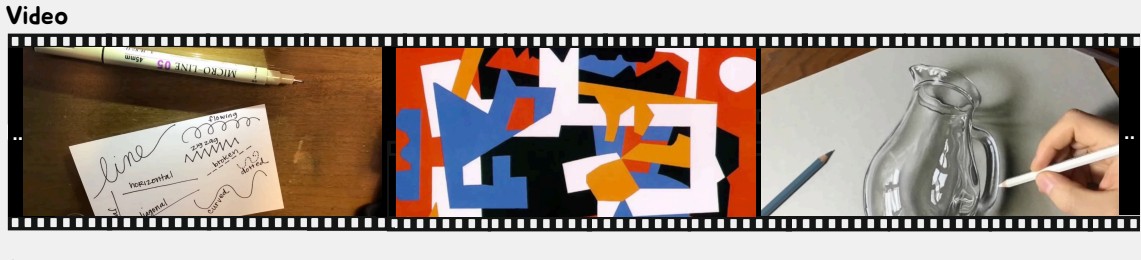

**Video**

**Question**

Which of the following elements of art are most prominently featured in this artwork?

**A. Line** B. Shape
C. Form D. Value
E. Color F. Texture
G. Balance H. Space
I. Contrast. J. Proportion

| Video-R1 | Ours |
|---|---|

**Video-R1**

The artwork in the video showcases various elements of art, but let's break down which one stands out the most.
1. Line: There are numerous lines used in the drawings, especially in the abstract and geometric shapes. The lines are varied in direction and style, contributing significantly to the overall composition.
2. Shape: Shapes are also prominent, with both geometric and organic forms present. The use of shapes to create patterns and structures is evident.
3. Form: While there are some three-dimensional elements like the cube and the sphere, they are not the main focus. The emphasis is more on two-dimensional elements rather than creating a sense of volume through form.
4. Value: The use of value (light and shadow) is minimal compared to other elements. The colors are bright and flat, without much shading.
5. Color: Color is present, but it is not the most dominant element. The colors are used to fill shapes and add contrast, but they do not create a strong color palette.
6. Texture: There is no significant texture in the artwork. The surfaces are smooth and flat.
7. Balance: The balance is somewhat achieved, but it is not the most striking feature. The arrangement of elements is somewhat balanced, but it is not the primary focus.
8. Space: The concept of space is present, but it is not the most prominent. The artwork does not explore depth or negative space extensively.
9. Contrast: There is some contrast, but it is not the most dominant element. The colors and lines provide some contrast, but it is not the main focus.
10. Proportion: The proportions are somewhat consistent, but they are not the most striking feature.

Given the analysis, the most prominently featured elements are line and shape. However, since the question asks for the most prominent, and considering the variety of lines and shapes, the answer leans towards:

B. Shape

Final answer: **B. Shape**

*Relies on text*

**Ours**

To determine which elements of art are most prominently featured in the artwork, let's analyze the video content step by step:
1. Line: The video shows a hand drawing various types of lines such as horizontal, diagonal, curved, and dotted. This suggests that line is a significant element in the artwork.
2. Shape: The video also shows the drawing of different shapes like triangles, circles, and squares, indicating that shape is another prominent element.
3. Form: The video briefly shows a 3D form, which suggests that form is also an element being discussed.
4. Value: The video mentions value, which refers to the lightness or darkness of a color, and shows examples of how value can be used in art.
5. Color: The video includes a color wheel and discusses primary colors, indicating that color is a key element.
6. Texture: The video briefly shows a textured surface, suggesting that texture is also an element being discussed.
7. Balance: The video does not focus on balance as a primary element.
8. Space: The video mentions space, including positive and negative space, indicating that space is a significant element.

Given the content of the video, the elements that are most prominently featured are likely to be those that are discussed and demonstrated in the video. Based on the video content, the most prominent elements are: Line, Shape, Color, Texture, Space

However, since the question asks for the most prominently featured elements, and considering the video content, the most prominent elements are likely to be: A. Line, B. Shape, E. Color, F. Texture, H. Space

The most comprehensive answer would be: A. Line, B. Shape, E. Color, F. Texture, H. Space

But if we need to choose the most prominent one based on the video content, the answer would be:

A. Line

So, the correct answer is:

**A. Line**

*Correct visual reference*

☐ : Over-reliance on text ☐ : Correct visual reference

Figure 4: Qualitative comparison of reasoning paths on a VideoMMMU art analysis question. Given a video demonstrating visual art elements, the model must identify which elements are shown. VIDGROUND (right) references specific visual elements observed in the video frames, such as lines, shapes, and colors (blue boxes), leading to the correct answer. In contrast, Video-R1 (Feng et al., 2025) (left) analyzes artistic concepts abstractly without grounding in the actual video content (red boxes), arriving at the wrong answer. This illustrates how post-training on visually grounded data encourages models to attend to visual evidence rather than relying on linguistic priors.

can be answered using the question text alone, with the strongest models exceeding 50%. This was not just an issue for a single frontier model, but rather a consistent trend across many leading VLMs, and presents a serious issue for measuring video understanding progress. We found a similar trend in the composition of post-training data for video understanding. Over 30% of the data in one of the most popular post-training datasets, Video-R1-260K (Feng et al., 2025), was also answerable using text alone. Guided by these observations, we developed a simple yet effective post-training strategy, VIDGROUND, for improving video understanding in VLMs: using only visually grounded questions for post-training. In combination with a simple RL-based post-training algorithm, this strategy outperforms five strong baselines when measured on both visually grounded evaluation splits as well as standard benchmark performance. Our approach also provides notable benefits in training data efficiency, achieving stronger performance with considerably less data. Furthermore, models post-trained on visually grounded data exhibit more consistent frame-scaling behavior, continuing to improve as more visual frames are provided, whereas baselines trained on unfiltered data plateau or degrade (Table 3). Overall, our findings highlight the importance of curating post-training data that truly requires visual reasoning, offering a simple yet powerful direction for building more robust and visually grounded VLMs.

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

---

**Appendix Overview**

---

---

In the appendix, we provide additional empirical evidence and implementation details supporting our main findings. We first analyze multi-model agreement on text-only answerability detection and compare alternative data curation strategies of varying strictness (§A). We then demonstrate cross-task generalization in §B, showing that visually grounded post-training does not degrade image QA performance. We present a comprehensive text-only answerability analysis across video understanding datasets (§C), evaluating a broad range of frontier models to demonstrate the pervasive nature of linguistic biases in video benchmarks. We provide implementation details including training configurations, evaluation setup, and computational resources in §D. Finally, we present additional qualitative analyses (§E) comparing reasoning paths between Video-R1 (Feng et al., 2025) and VIDGROUND, illustrating how video-dependent post-training data leads to models that frequently refer to video content rather than relying on linguistic shortcuts and exhibit stronger visually grounded reasoning abilities.

## A  Multi-model agreement analysis

### A.1  Multi-model agreement

To validate the robustness of text-only answerability detection, we evaluate three frontier models—GPT-5-mini (OpenAI, 2025), Qwen2.5-VL-7B (Bai et al., 2025), and Gemini-3.1-Pro (Google, 2026)—on Video-R1-260K (Feng et al., 2025) in text-only mode. Figure 5 shows the overlap of visually grounded (VG) questions across models.

Key findings: (1) The three models show strong agreement on which questions require visual input. GPT-5-mini (OpenAI, 2025) cannot answer 181,710 questions (69.1%) correctly without video, Qwen2.5-VL-7B (Bai et al., 2025) cannot answer 198,652 (75.5%), and their intersection (both models fail) contains 154,860 (58.9%). All three models fail on the same 145,486 questions (55.3%), forming a robust core of visually grounded data. (2) Inter-model agreement on VG questions is high (Jaccard index 68.7%), confirming that questions requiring visual understanding are consistently identified across diverse model architectures, supporting the robustness of our data curation approach.

### A.2  Alternative data curation strategies

We investigate whether multi-model consensus data curation improves upon our single-model approach. We evaluate three frontier models on all 263,071 samples from Video-R1-260K (Feng et al., 2025) in text-only mode (no visual input):

1. **GPT-5-mini** (OpenAI, 2025): Single-pass text-only evaluation on all samples. The model answers 81,361 questions (30.9%) correctly without video, leaving 181,710 VG questions (69.1%).

2. **Qwen2.5-VL-7B** (Bai et al., 2025): For MCQ questions, we employ circular evaluation (Fang et al., 2024) with option permutation to mitigate positional bias—a question is considered VG unless the model answers correctly under *all* permutations. For non-MCQ questions, we use Pass@10 sampling (10 independent responses; VG if none correct). With 2-permutation circular evaluation, 198,652 questions (75.5%) remain VG.

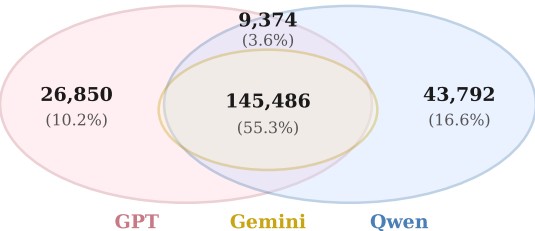

Figure 5: Multi-model agreement analysis on Video-R1-260K (Feng et al., 2025). Venn diagram showing the overlap of visually grounded (VG) questions—those that each model cannot answer correctly in text-only mode (no visual input)—across three frontier models: GPT-5-mini (OpenAI, 2025), Qwen2.5-VL-7B (Bai et al., 2025), and Gemini-3.1-Pro (Google, 2026). GPT cannot answer 181,710 questions (69.1%) correctly without video, Qwen cannot answer 198,652 (75.5%), and their intersection (GPT ∩ Qwen) contains 154,860 questions (58.9%) that neither model can solve without visual input. All three models fail on the same 145,486 questions (55.3%), forming a robust core of visually grounded data. Inter-model agreement on VG questions is high (Jaccard index 68.7%), confirming that questions requiring visual understanding are consistently identified across diverse architectures.

3. **Gemini-3.1-Pro** (Google, 2026): For MCQ questions, 3-permutation circular evaluation. For non-MCQ questions, direct text-only evaluation. Samples not covered by Gemini evaluation (∼2K) are conservatively classified as visually grounded (VG).

A question is retained as visually grounded (VG) only if fewer than 2 models can answer it correctly in text-only mode. This soft consensus threshold balances curation quality with data retention—requiring all three models to fail (<3 correct) would retain too many borderline questions, while requiring all models to fail (<1 correct) would discard too aggressively.

**Model selection rationale.** We select these three models to span diverse architectures and capability levels: GPT-5-mini as a strong closed-source model that serves as our primary single-model filter, Qwen2.5-VL-7B as the on-policy training model (see below), and Gemini-3.1-Pro as the strongest available model to maximize detection of questions answerable without visual input. Fig. 6 illustrates the multi-model data curation pipeline.

**Data curation variants.** We compare three approaches of increasing strictness:

- **VidGround (single-model, 181K)**: Our primary approach (§4.2), using GPT-5-mini only. Retains 181,710 samples (69.1%).

- **VidGround-M1 (161K)**: Soft multi-model data curation strategy with 2-permutation circular evaluation for Qwen2.5-VL-7B. Retains 160,837 samples (61.1%).

- **VidGround-M2 (148K)**: Stricter variant using 4-permutation circular evaluation for Qwen2.5-VL-7B, reducing false negatives from positional bias. Retains 147,850 samples (56.2%).

**On-policy data curation with Qwen2.5-VL-7B.** Notably, Qwen2.5-VL-7B serves a dual role: it is both a curation model and the base model for post-training. This on-policy curation is motivated by the insight that questions the training model itself can answer without visual input are precisely those that reinforce linguistic shortcuts during training. If Qwen2.5-VL-7B can solve a question through text alone, training on that question is unlikely to improve its visual grounding.

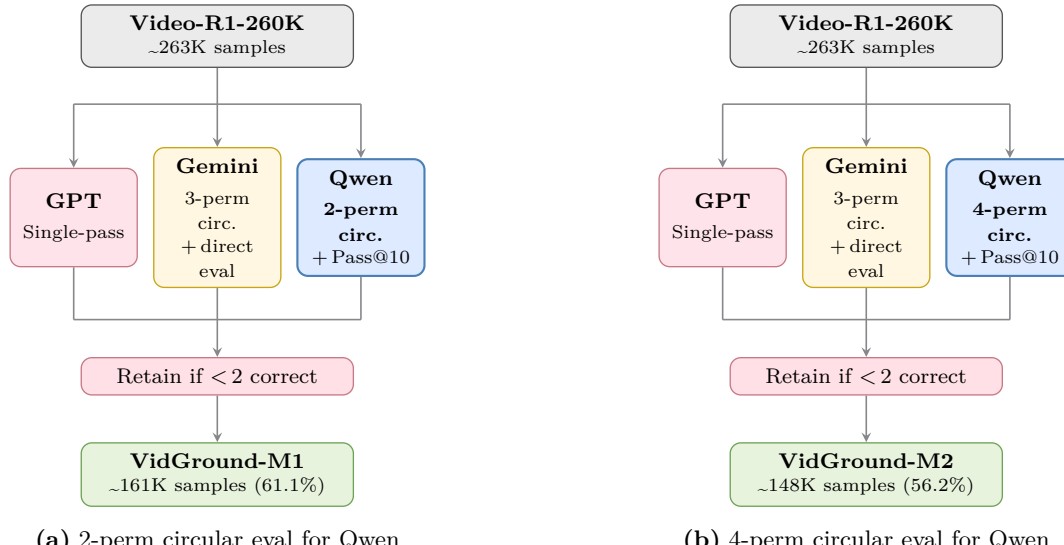

**(a)** 2-perm circular eval for Qwen         **(b)** 4-perm circular eval for Qwen

Figure 6: Multi-model data curation pipelines for VIDGROUND-M1 and VIDGROUND-M2 variants: GPT-5-mini (OpenAI, 2025), Qwen2.5-VL-7B (Bai et al., 2025), and Gemini-3.1-Pro (Google, 2026). Both pipelines evaluate all ∼263K samples from Video-R1-260K (Feng et al., 2025) in text-only mode. GPT uses single-pass evaluation, Qwen uses circular evaluation with option permutation for MCQ + Pass@10 for open-ended, and Gemini uses 3-permutation circular evaluation for MCQ + direct text evaluation for open-ended. A question is retained as VG only if fewer than 2 models can answer it correctly without visual input. The key difference is the number of circular evaluation permutations for Qwen: (a) VIDGROUND-M1 uses 2-permutation evaluation, retaining ∼161K samples (61.1%); (b) VIDGROUND-M2 uses stricter 4-permutation evaluation, retaining ∼148K samples (56.2%). Highlighted boxes indicate the differing component.

**Results and analysis.** Table 4 compares our three data curation strategies against the Qwen2.5-VL-7B base model and Video-R1 (Feng et al., 2025) (trained on the unfiltered 263K set). All models are post-trained with GRPO on Qwen2.5-VL-7B. Notably, all three VIDGROUND variants improve over both baselines: Video-R1 *degrades* the base model (−2.6 and −2.2 Avg. Full at 16 and 32 frames, respectively), whereas every VG-curated variant yields clear gains, confirming that retaining only visually grounded questions is essential for effective video reasoning training. Among the three variants, VIDGROUND-M1 achieves the highest average accuracy on the full benchmarks, slightly outperforming VIDGROUND (+0.2 at 16 frames, +0.4 at 32 frames). On the VG question subset, VIDGROUND-M1 also leads at 16 frames (45.9 vs. 45.2) while VIDGROUND narrowly leads at 32 frames (47.6 vs. 47.5). However, VIDGROUND-M2 (stricter curation) underperforms VIDGROUND-M1 on both full and VG metrics despite applying stricter retention criteria, suggesting that overly aggressive curation reduces training data diversity without proportional quality gains. Importantly, VIDGROUND with single-model curation remains highly competitive while being substantially simpler—requiring only one model evaluation pass rather than three—making it the most practical choice for large-scale data curation.

# B  Cross-task generalization

To verify that visually grounded post-training does not degrade performance on non-video tasks, we evaluate VIDGROUND on image QA benchmarks. As shown in Table 5, VIDGROUND improves over the Qwen2.5-VL-7B base model on all three benchmarks (MME: 648.9 vs. 624.3, MMMU: 58.7 vs. 56.7, MMBench: 84.5 vs. 84.2). These results demonstrate that curating post-training data for visual grounding does not narrow the training distribution in ways that harm non-video capabilities.

Table 4: Comparison of data curation strategies on video understanding benchmarks. VIDGROUND uses single-model curation (GPT-5-mini, 181K samples). VIDGROUND-M1 and VIDGROUND-M2 use progressively stricter multi-model consensus curation ($\geq 2$ models agree). All models post-trained with GRPO on Qwen2.5-VL-7B. Avg. columns report mean accuracy on the full benchmarks (Full) and on VG question subsets—questions that require video to answer. Deltas show improvement over Qwen2.5-VL-7B: $(+x)$ and $(-x)$. Bold indicates best; highlighted rows are ours.

| Frames | Method | VideoMME | VideoMMMU | MMVU | Avg. | |
| --- | --- | --- | --- | --- | --- | --- |
| | | | | | Full | VG |
| 16 | Qwen2.5-VL-7B | 58.2 | 45.0 | 60.5 | 54.6 | 42.9 |
| | *Video-R1* (Full, 263K) | 56.9 | 44.7 | 54.5 | 52.0 $(-2.6)$ | 41.7 $(-1.2)$ |
| | **VidGround** (VG, 181K) | **58.7** | 47.4 | 64.2 | 56. $(+2.2)$ | 45. $(+2.3)$ |
| | VIDGROUND-M1 (161K) | 58.5 | **48.0** | **64.4** | 57. $(+2.4)$ | 45. $(+3.0)$ |
| | VIDGROUND-M2 (148K) | 57.7 | 47.0 | 62.5 | 55. $(+1.1)$ | 43. $(+0.9)$ |
| 32 | Qwen2.5-VL-7B | 60.7 | 45.4 | 62.3 | 56.1 | 44.4 |
| | *Video-R1* (Full, 263K) | 60.2 | 45.4 | 56.2 | 53.9 $(-2.2)$ | 43.1 $(-1.3)$ |
| | **VidGround** (VG, 181K) | 61.5 | 48.3 | **65.8** | 58. $(+2.4)$ | 47. $(+3.2)$ |
| | VIDGROUND-M1 (161K) | **62.1** | **50.9** | 63.7 | 58. $(+2.8)$ | 47. $(+3.1)$ |
| | VIDGROUND-M2 (148K) | 61.4 | 50.4 | 62.8 | 58. $(+2.1)$ | 46. $(+2.2)$ |

Table 5: Performance on image QA benchmarks. VIDGROUND maintains or improves performance on non-video tasks compared to the Qwen2.5-VL-7B base model, demonstrating that visually grounded post-training does not harm cross-task generalization. Deltas show improvement over Qwen2.5-VL-7B: $(+x)$. Bold indicates best; highlighted row is ours.

| Method | MME | MMMU | MMBench |
| --- | --- | --- | --- |
| Qwen2.5-VL-7B | 624.3 | 56.7 | 84.2 |
| **VidGround** | **648.9** $(+24.6)$ | **58.7** $(+2.0)$ | **84.5** $(+0.3)$ |

## C  Text-only answerability analysis across video understanding datasets

To provide a comprehensive view of linguistic biases in video understanding benchmarks, we extend our text-only evaluation in Table 1 to a broader range of frontier models and report their performance on three video benchmarks: VideoMME (Fu et al., 2025a), VideoMMMU (Hu et al., 2025), and MMVU (Zhao et al., 2025). Table 6 presents results for 17 frontier models spanning closed-source VLMs including GPT-4o (Hurst et al., 2024), GPT-5-mini and GPT-5 (OpenAI, 2025), the Gemini (Comanici et al., 2025; Google, 2026) family, and Claude (Anthropic, 2025; 2026); open-source VLMs from the Qwen2.5-VL (Bai et al., 2025) family; and text-based large language models (LLMs) including DeepSeek-V3 and GPT-OSS (Agarwal et al., 2025) series. Notably, the LLMs evaluated have no visual capabilities and have never been trained on image or video data, making them well suited for assessing the extent to which video understanding benchmarks can be solved through linguistic reasoning alone.

### C.1  Key findings

**All VLMs and LLMs substantially exceed chance performance.** Every evaluated VLM and LLM achieves accuracy more than 20 points above random chance, with the strongest model, Gemini-3.1-Pro (Google, 2026), reaching +42.7 points above random. This universal trend across diverse model architectures suggests that linguistic shortcuts are not artifacts of specific model designs. Instead, they stem from the pervasive presence of text-only answerable questions in both evaluation benchmarks and the training data and are further exacerbated by potential contamination of training data with benchmark content when developing these models. When VLMs are trained on datasets containing substantial proportions of

Table 6: Extended text-only answerability across video understanding benchmarks for 17 frontier models spanning closed-source VLMs, open-source VLMs, and text-only LLMs. Each model receives only the question text and answer options—no video input. $(+x)$ denotes improvement over random chance. All model families achieve accuracy >20 points above random, confirming pervasive linguistic bias. Bold indicates best.

| Model | VideoMME | VideoMMMU | MMVU | Avg. |
|---|---|---|---|---|
| Random Choice | 25.0 | 9.8 | 19.8 | 18.2 |
| *Closed-source VLMs* | | | | |
| GPT-4o (Hurst et al., 2024) | 47.0 (+22.0) | 38.6 (+28.8) | 46.6 (+26.8) | 44.1 (+25.9) |
| GPT-5-mini (OpenAI, 2025) | 45.2 (+20.2) | 37.9 (+28.1) | 53.3 (+33.5) | 45.5 (+27.3) |
| GPT-5 (OpenAI, 2025) | 48.2 (+23.2) | 41.0 (+31.2) | 57.1 (+37.3) | 48.8 (+30.6) |
| Gemini-2.0-Flash (Comanici et al., 2025) | 43.1 (+18.1) | 43.6 (+33.8) | 53.4 (+33.6) | 46.7 (+28.5) |
| Gemini-2.5-Flash-Lite (Comanici et al., 2025) | 38.7 (+13.7) | 39.6 (+29.8) | 51.0 (+31.2) | 43.1 (+24.9) |
| Gemini-2.5-Flash (Comanici et al., 2025) | 49.6 (+24.6) | 48.1 (+38.3) | 55.5 (+35.7) | 51.1 (+32.9) |
| Gemini-2.5-Pro (Comanici et al., 2025) | 53.3 (+28.3) | 52.7 (+42.9) | 60.6 (+40.8) | 55.5 (+37.3) |
| Gemini-3.1-Pro (Google, 2026) | **58.2** (+33.2) | **61.1** (+51.3) | **63.4** (+43.6) | **60.9 (+42.7)** |
| Claude-Sonnet-4.5 (Anthropic, 2025) | 47.7 (+22.7) | 44.3 (+34.5) | 55.4 (+35.6) | 49.1 (+30.9) |
| Claude-Opus-4.6 (Anthropic, 2026) | 51.3 (+26.3) | 52.7 (+42.9) | 61.0 (+41.2) | 55.0 (+36.8) |
| *Open-source VLMs* | | | | |
| Qwen2.5-VL-7B (Bai et al., 2025) | 39.2 (+14.2) | 29.8 (+20.0) | 48.6 (+28.8) | 39.2 (+21.0) |
| Qwen2.5-VL-32B (Bai et al., 2025) | 41.3 (+16.3) | 42.0 (+32.2) | 54.7 (+34.9) | 46.0 (+27.8) |
| Qwen2.5-VL-72B (Bai et al., 2025) | 44.3 (+19.3) | 43.9 (+34.1) | 56.0 (+36.2) | 48.1 (+29.9) |
| *Open-source LLMs* | | | | |
| DeepSeek-V3.1-Terminus (Liu et al., 2025) | 39.0 (+14.0) | 26.8 (+17.0) | 52.6 (+32.8) | 39.5 (+21.3) |
| DeepSeek-V3.2-Exp (Liu et al., 2025) | 39.6 (+14.6) | 23.8 (+14.0) | 53.9 (+34.1) | 39.1 (+20.9) |
| GPT-OSS-20B | 40.4 (+15.4) | 37.8 (+28.0) | 50.6 (+30.8) | 42.9 (+24.7) |
| GPT-OSS-120B | 45.0 (+20.0) | 37.9 (+28.1) | 53.3 (+33.5) | 45.4 (+27.2) |

linguistically biased or contaminated examples, they inevitably learn to exploit textual patterns and rely on their pretrained knowledge rather than developing robust visual grounding capabilities. This observation motivates our VIDGROUND approach: by filtering training data to retain only visually-dependent questions, we can mitigate the linguistic biases that current training paradigms inadvertently amplify.

**More than half of the benchmark questions require no video.** Gemini-3.1-Pro achieves 58–64% accuracy across all three benchmarks without any visual information (VideoMME: 58.2%, VideoMMMU: 61.1%, MMVU: 63.4%), and even Gemini-2.5-Pro reaches approximately 50% or higher on each benchmark (53.3%, 52.7%, 60.6% respectively). GPT-5 also achieves 57.1% on MMVU without any visual input. These results show that half of the questions in these widely used video benchmarks can be answered correctly using text alone, casting doubt on their validity as measures of genuine video understanding.

Notably, VideoMME (Fu et al., 2025a) reports that "Gemini-1.5-Pro achieves less than 15% accuracy in the text-only setup"; however, our experiments show Gemini-1.5-Pro achieves 41.4% on VideoMME in text-only mode, and Gemini-2.5-Flash reaches 49.6%. We find that this discrepancy likely stems from the sensitivity of VLMs to prompts: with slight changes to the prompt, models that initially refuse to answer will readily produce responses in the text-only setting. Further details are provided in §D.4.

**Performance gains from model scaling primarily reflect stronger language understanding.** Consistent performance improvements are observed as model capacity increases within each model family: GPT-5 outperforms GPT-5-mini (+3.3 average points), Gemini-2.5-Flash surpasses Gemini-2.5-Flash-Lite (+8.0 points), and Qwen2.5-VL-72B exceeds both the 32B (+2.1 points) and 7B variants (+8.9 points). Critically, these gains persist in the text-only setting, indicating that VLM scaling benefits stem primarily from enhanced linguistic reasoning rather than improved visual grounding. This pattern holds for both closed-source and open-source models, revealing that apparent progress on video benchmarks largely reflects stronger language capabilities, not visually grounded video understanding.

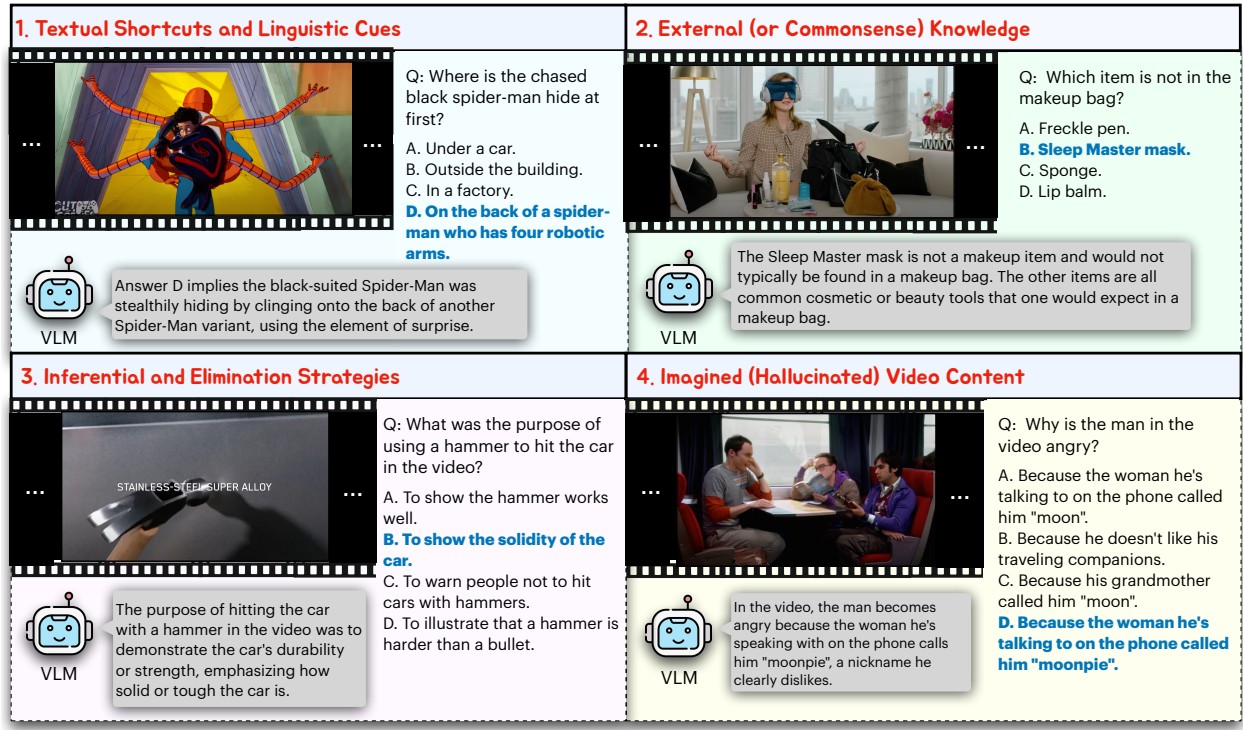

Figure 7: Additional examples of text-only answerable (TA) questions from VideoMME (Fu et al., 2025a). VLMs can exploit the same four categories of linguistic biases identified in Figure 2a to answer benchmark questions without visual grounding. VLM responses from GPT-4o (Hurst et al., 2024).

**Large language models rival or exceed vision-language models.** Remarkably, text-only LLMs, which have never been exposed to visual data during training, achieve competitive or superior performance compared to VLMs. GPT-OSS-120B (45.4% average) outperforms GPT-4o (44.1%), and DeepSeek-V3.2-Exp achieves 53.9% on MMVU, exceeding GPT-5-mini (53.3%) and rivaling several VLMs. Surprisingly, when comparing LLMs in text-only mode against VLMs with full video access, LLMs remain competitive or superior: GPT-OSS-120B reaches 45.0% on VideoMME without any visual input, matching or exceeding VLMs such as Video-LLaVA (Lin et al., 2024a) (39.9%) and Chat-UniVi-V1.5 (Jin et al., 2024) (40.6%) with video input. These results demonstrate that current video benchmarks are solvable primarily through linguistic shortcuts, commonsense reasoning, and world knowledge rather than visual comprehension, as LLMs correctly answer approximately 39–45% of questions on average using language capabilities alone.

## C.2    Analysis of linguistic biases in video benchmarks

To validate the prevalence of linguistic biases in video understanding benchmarks, we analyze representative examples from VideoMME (Fu et al., 2025a) (Figure 7). These examples demonstrate how each of the four TA categories manifests in widely-used benchmarks, enabling models to bypass genuine video understanding.

**Textual shortcuts and linguistic cues.** In the Spider-Man question, options include generic locations ("under a car," "in a factory") alongside one highly specific choice: "on the back of a Spider-Man who has four robotic arms." This unusual specificity enables models to identify the correct answer through linguistic pattern matching, rewarding text-based reasoning over multimodal grounding.

**External knowledge.** When asked "Which item is not in the makeup bag?" with options including freckle pen, Sleep Master mask, sponge, and lip balm, models can identify the Sleep Master mask as the outlier

based solely on categorical knowledge, as it is not a typical cosmetic item. This requires no visual evidence of the bag's actual contents.

**Inferential and elimination strategies.** For "What was the purpose of using a hammer to hit the car?", options include implausible purposes alongside one reasonable explanation: demonstrating the car's solidity. Models can identify the correct answer through elimination of implausible options rather than observing the video content.

**Imagined (hallucinated) video content.** When asked "Why is the man in the video angry?", models might generate the correct answer "Because the woman called him moonpie" by hallucinating plausible conversation scenarios. Since the anger stems from dialogue rather than visual cues, such fortunate hallucinations do not reflect genuine video understanding.

These examples confirm that the four TA categories identified in training data (Figure 2a) also pervade evaluation benchmarks. The presence of such questions allows models to achieve high accuracy through linguistic shortcuts rather than genuine video reasoning.

## C.3 Quantifying text-only answerability via Pass@10 sampling

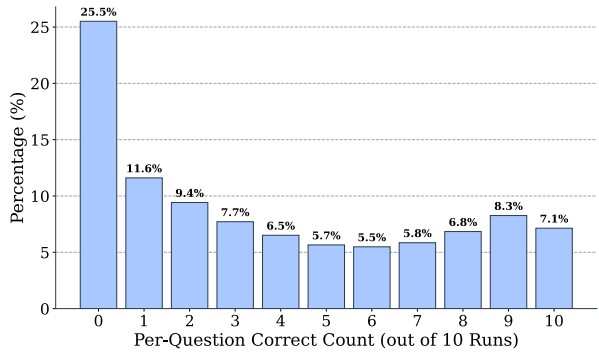
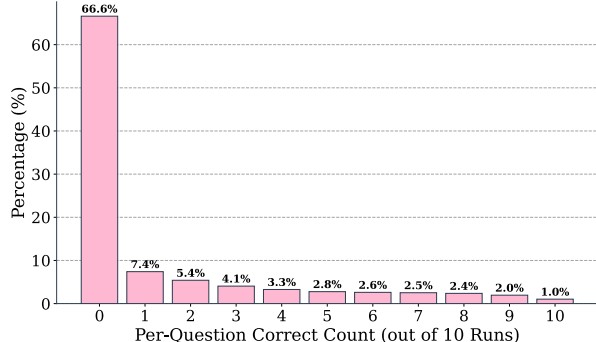

Figure 8: Pass@10 distribution for video questions in Video-R1-260K (Feng et al., 2025). Pass@10 measures the fraction of questions where the model produces at least one correct answer across 10 independent text-only samples. Qwen2.5-VL-7B (Bai et al., 2025) achieves Pass@10 > 0 on 74.5% of video questions without video input.

Figure 9: Pass@10 distribution for image questions in Video-R1-260K (Feng et al., 2025). Same setup as Figure 8. Qwen2.5-VL-7B (Bai et al., 2025) achieves Pass@10 > 0 on 33.4% of image questions without visual input.

To quantify text-only answerability in training data, we evaluate Qwen2.5-VL-7B (Bai et al., 2025) on all 263,071 instances of Video-R1-260K (Feng et al., 2025) without providing visual input. For each question, we generate 10 independent responses and record how many are correct. We then compute Pass@10 as the percentage of questions where at least one of the 10 responses is correct. As shown in Figures 8, 9, and 10, 74.5% of video questions (i.e., questions that require video to answer), 33.4% of image questions (i.e., questions that require images to answer), and 51.6% overall achieve Pass@10 > 0.

The distribution patterns reveal different characteristics across modalities. Video questions (Figure 8) show 25.5% never answered correctly and 7.1% answered correctly in all 10 runs, with relatively uniform distribution across intermediate values. This suggests video questions are more susceptible to linguistic shortcuts. In contrast, image questions (Figure 9) exhibit a skewed distribution with 66.6% never correct and only 1.0% always correct, indicating stronger visual dependency. The overall distribution (Figure 10) reflects these patterns with 48.4% never correct and 3.7% always correct.

We hypothesize that this disparity stems from differences in data construction costs. Video QA data is significantly more expensive and difficult to construct than image QA data, leading to widespread reliance

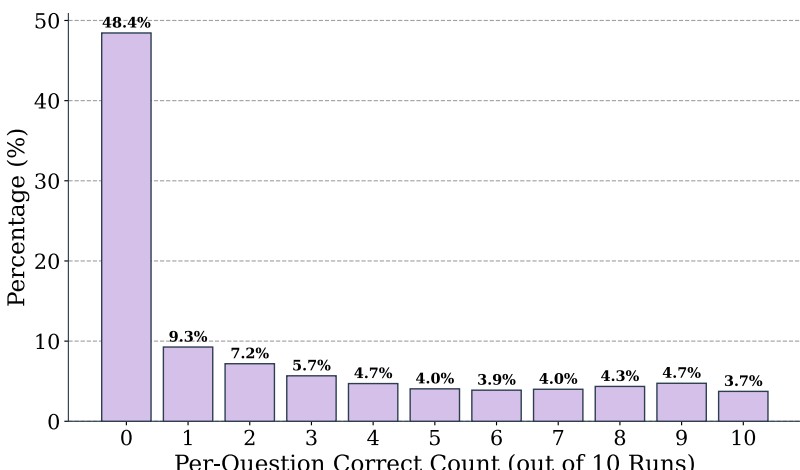

Figure 10: Overall Pass@10 distribution for Video-R1-260K (Feng et al., 2025). Same setup as Figure 8. 51.6% of all questions achieve Pass@10 > 0 by Qwen2.5-VL-7B (Bai et al., 2025) without visual information.

on LLM-generated question-answer pairs. This practice introduces severe linguistic biases, as models trained on LLM-generated questions inherit the text-based reasoning patterns of their generators. For instance, LLaVA-Video-178K (Zhang et al., 2024b), a widely-used training dataset, employs GPT-4o to generate question-answer pairs. While such LLM-generated data offers scalability, it systematically increases the proportion of TA (text-only answerable) questions, amplifying the linguistic bias problem revealed by our Pass@10 analysis.

### C.4 Implications for video understanding research

The results across diverse model families in Table 6 and the text-only answerability analysis in §C.1, §C.2, and §C.3 establish several key implications.

**Current benchmarks systematically overestimate progress.** When 30–50% (or more) of performance can be attributed to linguistic shortcuts rather than visual understanding, these benchmarks exaggerate actual progress in video comprehension. Reported improvements may largely reflect advances in language modeling rather than genuine progress in multimodal video comprehension.

**Visually grounded training data is essential.** The strong performance of VLMs on video benchmarks and training datasets in the text-only setting demonstrates that current training data introduces linguistic biases that undermine visual grounding. As shown in our ablation study (Table 3), training on the full dataset—which contains text-only answerable questions—leads to worse performance than training on visually grounded data alone, suggesting that linguistic shortcuts in training data undermine visual grounding. VIDGROUND mitigates this issue and promotes more faithful visual grounding.

**Periodic quality checks of video understanding benchmarks are required.** Future video understanding benchmark development should systematically filter text-only answerable questions using frontier language models, as demonstrated in our VG (visually grounded) benchmark subsets. Regular re-evaluation is necessary as language models continue to improve, rendering previously valid questions increasingly vulnerable to linguistic shortcuts.

# D    Implementation details

## D.1    Training configuration

We post-train Qwen2.5-VL-7B-Instruct (Bai et al., 2025) using the GRPO objective described in §4.1. Training is conducted on 8× NVIDIA H100 GPUs for 700 steps. We uniformly sample 16 frames from each video during training. Training hyperparameters are provided in Table 7. We employ TRL (von Werra et al., 2025) for GRPO implementation.

Table 7: Training configuration for VIDGROUND post-training. We apply GRPO (Shao et al., 2024) to Qwen2.5-VL-7B-Instruct (Bai et al., 2025) for one epoch on the VIDGROUND filtered dataset of 181K visually grounded samples from Video-R1-260K (Feng et al., 2025).

| Hyperparameter | Value |
|---|---|
| *Training Setup* | |
| model_name_or_path | Qwen/Qwen2.5-VL-7B-Instruct |
| max_prompt_length | 16384 |
| max_completion_length | 768 |
| per_device_train_batch_size | 1 |
| gradient_accumulation_steps | 1 |
| num_train_epochs | 1 |
| learning_rate | 1e-6 |
| lr_scheduler_type | cosine |
| weight_decay | 0.01 |
| max_grad_norm | 5 |
| bf16 | true |
| attn_implementation | flash_attention_2 |
| min_pixels | 3136 |
| max_pixels | 501760 |
| *GRPO Settings* | |
| temporal | true |
| len_control | true |
| $\beta$ | 0.04 |
| temperature | 1.0 |
| num_generations | 8 |
| $\epsilon$ | 0.2 |

## D.2    Evaluation setup

We evaluate all models on 4× NVIDIA L40S GPUs. Following standard practice, we uniformly sample 16, 32, and 64 frames per video when evaluating on video benchmarks to assess performance across different temporal resolutions.

### D.3 Text-only answerability filtering

To construct our VG (visually grounded) training dataset, we filter Video-R1-260K (Feng et al., 2025) by removing text-only answerable questions. We use GPT-5-mini (OpenAI, 2025) to evaluate each question-answer pair without visual content access, and questions answered correctly are classified as TA (text-only answerable) and removed. This filtering reduces the dataset from 263,071 to 181,710 samples (69.1% retention rate), removing 30.9% of linguistically biased examples.

### D.4 Prompt template for text-only evaluation

For text-only evaluation, we append additional prompts to the questions and options from each benchmark. Our default prompt provides minimal instruction (Figure 11). However, earlier Gemini models (e.g., Gemini-1.5-Pro and Gemini-2.0-Flash) often refuse to answer without video access. To obtain responses from these models, we use an enhanced prompt that explicitly prevents refusals (Figure 12).

---

**Prompt template for text-only evaluation**

Where is the main character in the video on holiday?

A. Sandcastle
B. Private mansion
C. Beachside
D. Music festival

Based on the question and options provided, use your knowledge and common sense to determine the most likely correct answer.

---

**Enhanced prompt template for text-only evaluation**

Which tool is not necessary to make a snow globe?

A. Distilled water    B. Scissors    C. Glitter    D. Super glue.

Based on the question and options provided, use your knowledge and common sense to determine the most likely correct answer. The video and subtitles are not available. Based only on the question and options, you must choose the most plausible answer. If unsure, make your best guess. Do not refuse to answer.

---

Figure 11: Default prompt template. Red text indicates the instructions added for text-only evaluation.

Figure 12: Enhanced prompt template. Red text indicates the instructions added to prevent refusals during text-only evaluation.

These different prompting results reveal a critical issue in Video-MME's data quality assessment methodology. Video-MME (Fu et al., 2025a) reports that *"Gemini 1.5 Pro achieves less than 15% accuracy in the text-only setup, underscoring the robustness of the video content-based requirement."* However, this low accuracy results from model refusal rather than genuine inability to answer. With our enhanced prompt that prevents refusals, Gemini-1.5-Pro achieves substantially higher text-only accuracy (41.4%), demonstrating that the benchmark contains significantly more text-only answerable questions than Video-MME reported. This finding highlights that evaluation protocols must carefully account for model refusal behavior and sensitivity to prompts when validating benchmark quality.

## E Additional qualitative analysis

We provide additional qualitative comparison examples in addition to Figure 4 to further demonstrate the behavioral differences between Video-R1 (Feng et al., 2025) and VIDGROUND. After manually inspecting reasoning chains across multiple instances, we identify a consistent pattern: VIDGROUND systematically grounds its reasoning in video content, whereas Video-R1 primarily relies on text-based analysis of the question and options without referencing visual context.

### E.1 Visually grounded reasoning vs. text-based reasoning

Across all examples (Figures 13–20), we make the following observations.

**VidGround consistently anchors reasoning in video content.** Most responses from VIDGROUND begin by establishing what information the video provides. For example:

- "Given the content of the video, the elements that are most prominently featured..." (Figure 13)

- "Since the video is likely discussing the pelvis and the red dots are..." (Figure 16)

- "Given the context of the video, which focuses on eosinophils..." (Figure 18)

- "The video is about influence lines in structural engineering..." (Figure 20)

Such reasoning paths relying on video content ensure responses are grounded in observed visual evidence rather than linguistic priors.

**Video-R1 relies more on text-based analysis and prior knowledge.** In contrast, Video-R1 typically begins with generic problem-solving templates and proceeds to analyze the question and options through:

- Textual shortcuts and linguistic cues ("Based on the information provided in the question and the table..." in Figure 17)

- General domain knowledge ("Tau is a microtubule-associated protein..." in Figure 18)

- Elimination strategies ("This option is too broad..." in Figure 20)

### E.2 Beyond accuracy metrics: identical answers through different reasoning paths

Figure 20 provides evidence that our filtering approach changes model behavior fundamentally, not merely accuracy. Both models correctly answer the structural engineering question about influence lines, yet their reasoning processes differ substantially.

Video-R1 begins by analyzing the answer options rather than referencing the video, indicating that it relies heavily on textual cues. For example, its reasoning chain interprets each choice purely through linguistic analysis, such as: "A. unit moving load with unchanged direction – This option suggests that the load is a unit moving load with its direction remaining constant. This is a common assumption in structural analysis where the load is considered as a unit force that moves along a specific path...". The model continues in this manner for all four options, using only textual reasoning and never engaging with the video content.

VIDGROUND explicitly grounds its analysis in the video, for example: "The video is about influence lines in structural engineering, which are used to determine the maximum values of various quantities...The key point here is that the influence line is derived by considering a unit load moving along the structure.". This response shows an understanding of the visual content before reaching the same conclusion.

This example indicates a limitation of accuracy-based evaluation. VLMs can produce correct answers through linguistic shortcuts by bypassing the understanding of the videos. Therefore, we argue that examining reasoning paths is important, as it exposes whether the model's intermediate thinking process relies on visual grounding or simply exploits text-based shortcuts.

### E.3 Summary

In summary, these qualitative comparisons (Figures 13–20) collectively demonstrate the advantages of VID-GROUND. Our method not only improves model performance significantly, but also guides the model's reasoning to be more visually grounded and less reliant on linguistic shortcuts. These examples provide solid evidence that VIDGROUND effectively mitigates the linguistic biases introduced by current video post-training datasets. Our findings highlight that visually grounded training data is crucial for training models that truly leverage video content rather than relying on text-based shortcuts.

**Video**

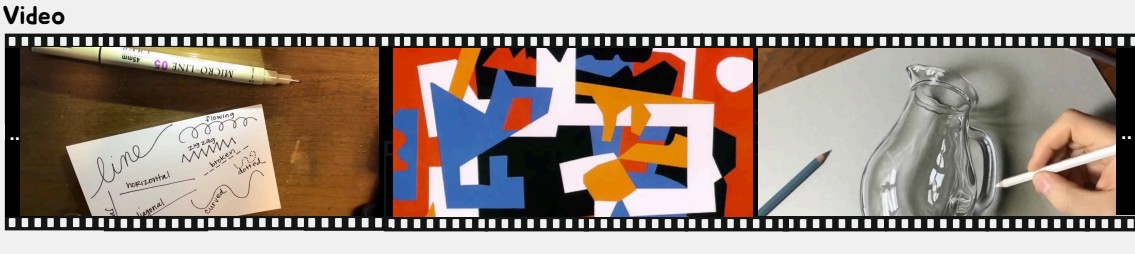

**Question**

Which of the following elements of art are most prominently featured in this artwork?

**A. Line**     B. Shape
C. Form     D. Value
E. Color     F. Texture
G. Balance     H. Space
I. Contrast.     J. Proportion

| Video-R1 | Ours |
|---|---|
| The artwork in the video showcases various elements of art, but let's break down which one stands out the most.
1. Line: There are numerous lines used in the drawings, especially in the abstract and geometric shapes. The lines are varied in direction and style, contributing significantly to the overall composition.
2. Shape: Shapes are also prominent, with both geometric and organic forms present. The use of shapes to create patterns and structures is evident.
3. Form: While there are some three-dimensional elements like the cube and the sphere, they are not the main focus. The emphasis is more on two-dimensional elements rather than creating a sense of volume through form.
4. Value: The use of value (light and shadow) is minimal compared to other elements. The colors are bright and flat, without much shading.
5. Color: Color is present, but it is not the most dominant element. The colors are used to fill shapes and add contrast, but they do not create a strong color palette.
6. Texture: There is no significant texture in the artwork. The surfaces are smooth and flat.
7. Balance: The balance is somewhat achieved, but it is not the most striking feature. The arrangement of elements is somewhat balanced, but it is not the primary focus.
8. Space: The concept of space is present, but it is not the most prominent. The artwork does not explore depth or negative space extensively.
9. Contrast: There is some contrast, but it is not the most dominant element. The colors and lines provide some contrast, but it is not the main focus.
10. Proportion: The proportions are somewhat consistent, but they are not the most striking feature.

Given the analysis, the most prominently featured elements are line and shape. However, *since the question asks for the most prominent, and considering the variety of lines and shapes, the answer leans towards:*

B. Shape

Final answer: **B. Shape**     **Relies on text** | To determine which elements of art are most prominently featured in the artwork, let's analyze the video content step by step:
1. Line: The video shows a hand drawing various types of lines such as horizontal, diagonal, curved, and dotted. This suggests that line is a significant element in the artwork.
2. Shape: The video also shows the drawing of different shapes like triangles, circles, and squares, indicating that shape is another prominent element.
3. Form: The video briefly shows a 3D form, which suggests that form is also an element being discussed.
4. Value: The video mentions value, which refers to the lightness or darkness of a color, and shows examples of how value can be used in art.
5. Color: The video includes a color wheel and discusses primary colors, indicating that color is a key element.
6. Texture: The video briefly shows a textured surface, suggesting that texture is also an element being discussed.
7. Balance: The video does not focus on balance as a primary element.
8. Space: The video mentions space, including positive and negative space, indicating that space is a significant element.

Given the content of the video, the elements that are most prominently featured are likely to be those that are discussed and demonstrated in the video. Based on the video content, the most prominent elements are: Line, Shape, Color, Texture, Space

However, since the question asks for the most prominently featured elements, and considering the video content, the most prominent elements are likely to be: A. Line, B. Shape, E. Color, F. Texture, H. Space

The most comprehensive answer would be: A. Line, B. Shape, E. Color, F. Texture, H. Space
But if we need to choose the most prominent one based on the video content, the answer would be:

A. Line     **Correct visual reference**

So, the correct answer is:

**A. Line** |

☐ : Over-reliance on text     ☐ : Correct visual reference

Figure 13: Art elements analysis question comparing VIDGROUND (trained on visually grounded data) with Video-R1 (Feng et al., 2025) (trained on the full, unfiltered Video-R1-260K dataset). VIDGROUND references the art elements most prominently featured in the video, while Video-R1 analyzes artistic concepts abstractly without considering the video content. Green highlights indicate correct reasoning grounded in video content; red highlights indicate text-based reasoning without visual grounding.

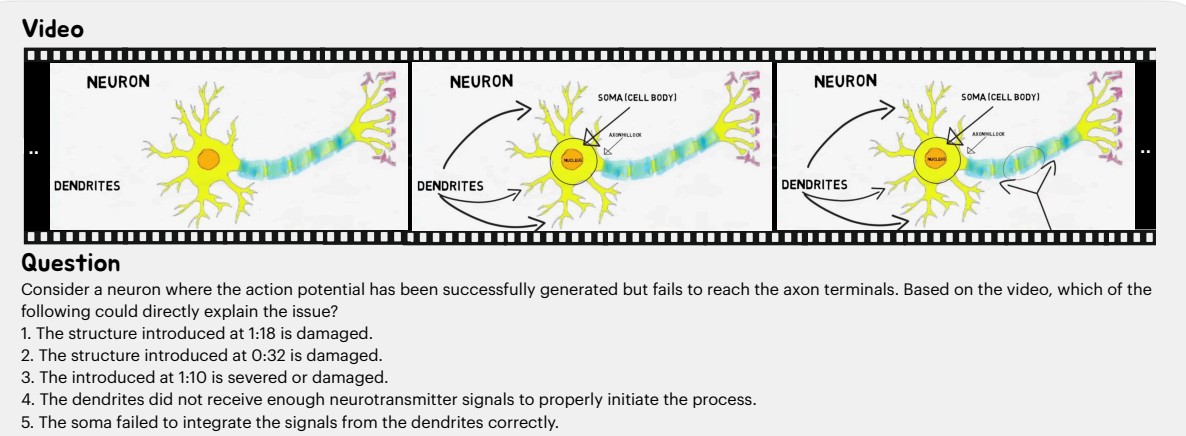

**Video**

**Question**

Consider a neuron where the action potential has been successfully generated but fails to reach the axon terminals. Based on the video, which of the following could directly explain the issue?
1. The structure introduced at 1:18 is damaged.
2. The structure introduced at 0:32 is damaged.
3. The introduced at 1:10 is severed or damaged.
4. The dendrites did not receive enough neurotransmitter signals to properly initiate the process.
5. The soma failed to integrate the signals from the dendrites correctly.

A. 1, 2, and 3 are correct   B. 2 and 4 are correct   **C. 1 and 3 are correct**   D. 3 and 5 are correct   E. Only 1 is correct
F. 2 and 5 are correct   G. "Only 4 is correct   H. 1 and 4 are correct   I. 1, 3, and 5 are correct   J. 2, 3, and 4 are correct

---

**Video-R1**

Let's break down the problem step by step.
First, we need to understand the components of a neuron and their roles:
- Dendrites: Receive signals from other neurons.
- Soma (Cell Body): Processes the signals received by the dendrites.
- Axon: Transmits the signal to the axon terminals.

The question asks about the scenario where an action potential is generated but fails to reach the axon terminals. This suggests that there might be an issue with the transmission of the signal from the soma to the axon terminals.

Now, let's consider each option:     **Relies on text**
1. The structure introduced at 1:18 is damaged. This refers to the axon hillock, which is a part of the axon. If the axon hillock is damaged, it could affect the transmission of the action potential.
2. The structure introduced at 0:32 is damaged. This refers to the dendrites. If the dendrites are damaged, they might not receive enough neurotransmitter signals to properly initiate the process.
3. The introduced at 1:10 is severed or damaged. This refers to the axon. If the axon is severed or damaged, it would prevent the action potential from reaching the axon terminals.
4. The dendrites did not receive enough neurotransmitter signals to properly initiate the process. This is a possibility, but it is more related to the dendrites' function rather than the axon's function.
5. The soma failed to integrate the signals from the dendrites correctly. This is also a possibility, but it is more related to the soma's function rather than the axon's function.

From this analysis, the most relevant options are 1, 3, and 5.
Therefore, the correct answer is I.
Final answer: I

---

**Ours**

The video explains the structure of a neuron and its components, including the dendrites, soma (cell body), axon hillock, axon, myelin, and axon terminals. The action potential is generated in the soma and travels down the axon to the axon terminals. If the action potential fails to reach the axon terminals, it could be due to damage to the axon or the axon terminals, which are the structures at the end of the axon where neurotransmitters are released.

**Correct visual reference**

Given the options:
1. The structure introduced at 1:18 is damaged. This refers to the axon hillock, which is the region where the action potential is generated.
2. The structure introduced at 0:32 is damaged. This refers to the soma (cell body), which is the main body of the neuron.
3. The introduced at 1:10 is severed or damaged. This refers to the axon, which is the long, thin projection of the neuron that transmits signals.
4. The dendrites did not receive enough neurotransmitter signals to properly initiate the process. This is not directly related to the failure of the action potential to reach the axon terminals.
5. The soma failed to integrate the signals from the dendrites correctly. This is not directly related to the failure of the action potential to reach the axon terminals.

The correct answer is:
**C. 1 and 3 are correct**

The action potential fails to reach the axon terminals if the axon hillock (1) is damaged or if the axon (3) is severed or damaged.

☐ : Over-reliance on text   ☐ : Correct visual reference

Figure 14: Psychology question. Same comparison setup as Figure 13. VIDGROUND grounds its reasoning in the neuron and its components introduced by the video, while Video-R1 (Feng et al., 2025) analyzes through general biology knowledge.

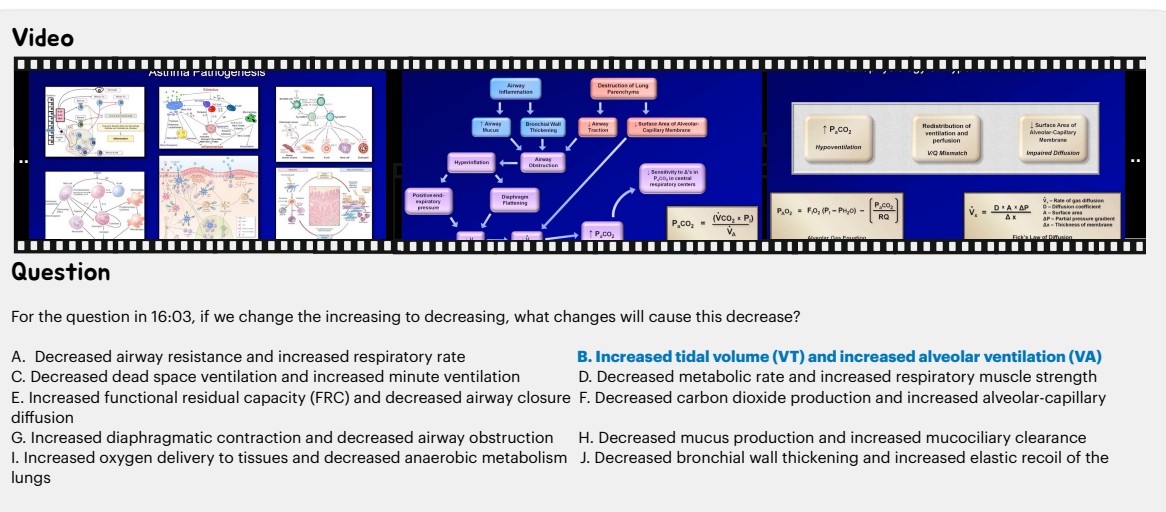

Figure 15: Medical science question. Same comparison setup as Figure 13. VIDGROUND locates the relevant content in the video about $PaCO_2$, while Video-R1 (Feng et al., 2025) evaluates options using linguistic priors.

**Video**

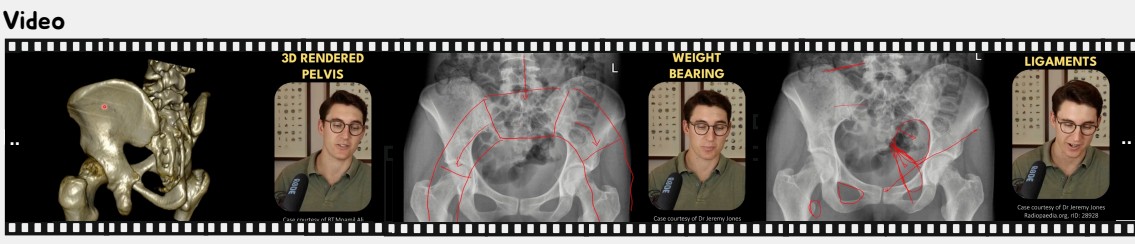

**Question**

Can you identify the abnormality on this plain film of the pelvis? <image 1>

A. Bone cyst
C. Osteoarthritis
**E. Resection of the pubic symphysis**
G. Hip dislocation
I. Paget's disease

B. Acute hip fracture
D. Surgical hardware
F. Bone tumor
H. Osteoporosis
J. Bone infection

**Video-R1**

Based on the provided options, let's analyze the image step by step:

**Relies on text**

1. Bone Cyst: A bone cyst typically appears as a well-defined, fluid-filled lesion in the bone. There is no clear evidence of such a lesion in the image.
2. Acute Hip Fracture: An acute hip fracture would show a clear break or discontinuity in the femoral neck or head. While there might be some irregularities, it's not immediately obvious that this is an acute fracture.
3. Osteoarthritis: Osteoarthritis usually shows degenerative changes with narrowing of joint spaces, osteophytes, and sclerosis. The image does not clearly depict these features.
4. Surgical Hardware: The image shows a metallic object within the pelvic region, which could be surgical hardware used for fixation or reconstruction.
5. Resection of the Pubic Symphysis: This would involve a significant change in the alignment and structure of the pubic bones, which is not evident here.
6. Bone Tumor: A bone tumor would show a mass effect with possible destruction of the bone. There is no clear evidence of a tumor in the image.
7. Hip Dislocation: While there might be some displacement, the image does not clearly show the typical signs of a dislocated hip.
8. Osteoporosis: Osteoporosis typically shows decreased bone density and increased risk of fractures. The image does not clearly depict these features.
9. Paget's Disease: Paget's disease can show irregular bone density and deformities. The image does not clearly depict these features.
10. Bone Infection: A bone infection would show signs of inflammation and possibly abscesses. There is no clear evidence of infection in the image.

Given the presence of a metallic object within the pelvic region, the most likely abnormality is surgical hardware.

Therefore, the correct answer is **D. Surgical hardware.**

**Ours**

The image provided is a plain film of the pelvis, and it shows a 3D rendered pelvis with red dots indicating specific areas of interest. The red dots are likely highlighting bony landmarks or areas of concern. Given the context of the video, which seems to be educational and focused on medical imaging, the most likely abnormality to be identified is a specific bony structure or a condition related to the pelvis.

**Correct visual reference**

However, without a clear visual of the red dots and their specific locations, it is challenging to definitively identify the abnormality. The options provided include various conditions such as bone cyst, acute hip fracture, osteoarthritis, surgical hardware, resection of the pubic symphysis, bone tumor, hip dislocation, osteoporosis, Paget's disease, and bone infection.

Since the video is likely discussing the pelvis and the red dots are pointing to specific bony landmarks, the most probable answer would be related to a bony abnormality. Among the options, the most common and relevant one would be:

E. Resection of the pubic symphysis

This is because the pubic symphysis is a common area of interest in pelvic imaging, and resection of this area would be a significant abnormality that would be highlighted in a medical educational video.

Therefore, the most likely answer is:
**E. Resection of the pubic symphysis**

☐: Over-reliance on text   ☐: Correct visual reference

Figure 16: Clinical medical imaging interpretation question. Same comparison setup as Figure 13. This instance includes a reference image as part of the question prompt. VIDGROUND refers to the video content about the pelvis and red dots pointing to specific bony landmarks, while Video-R1 (Feng et al., 2025) relies on prior knowledge to eliminate options without referencing the video.

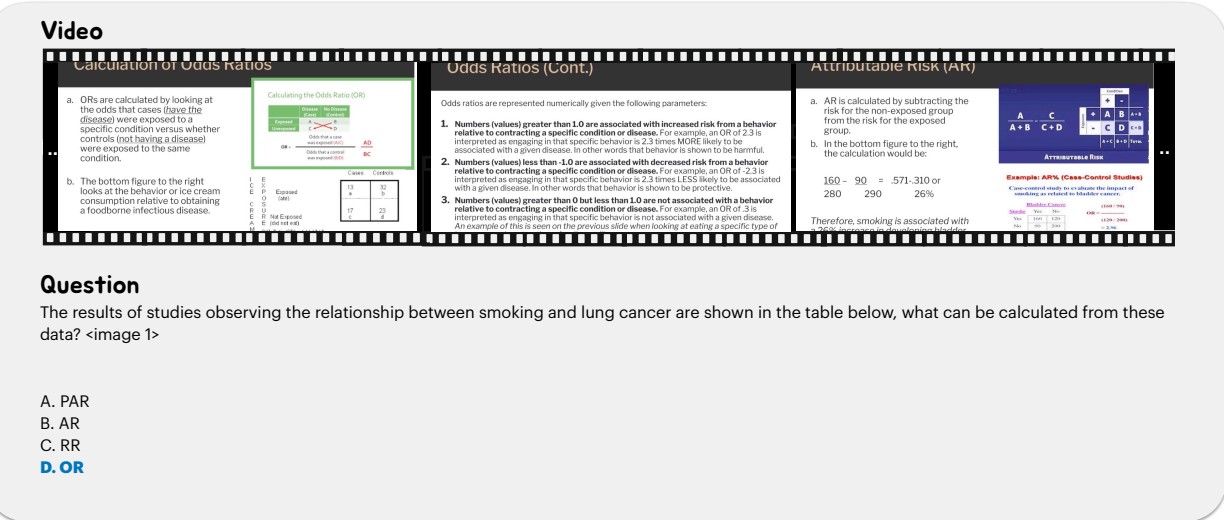

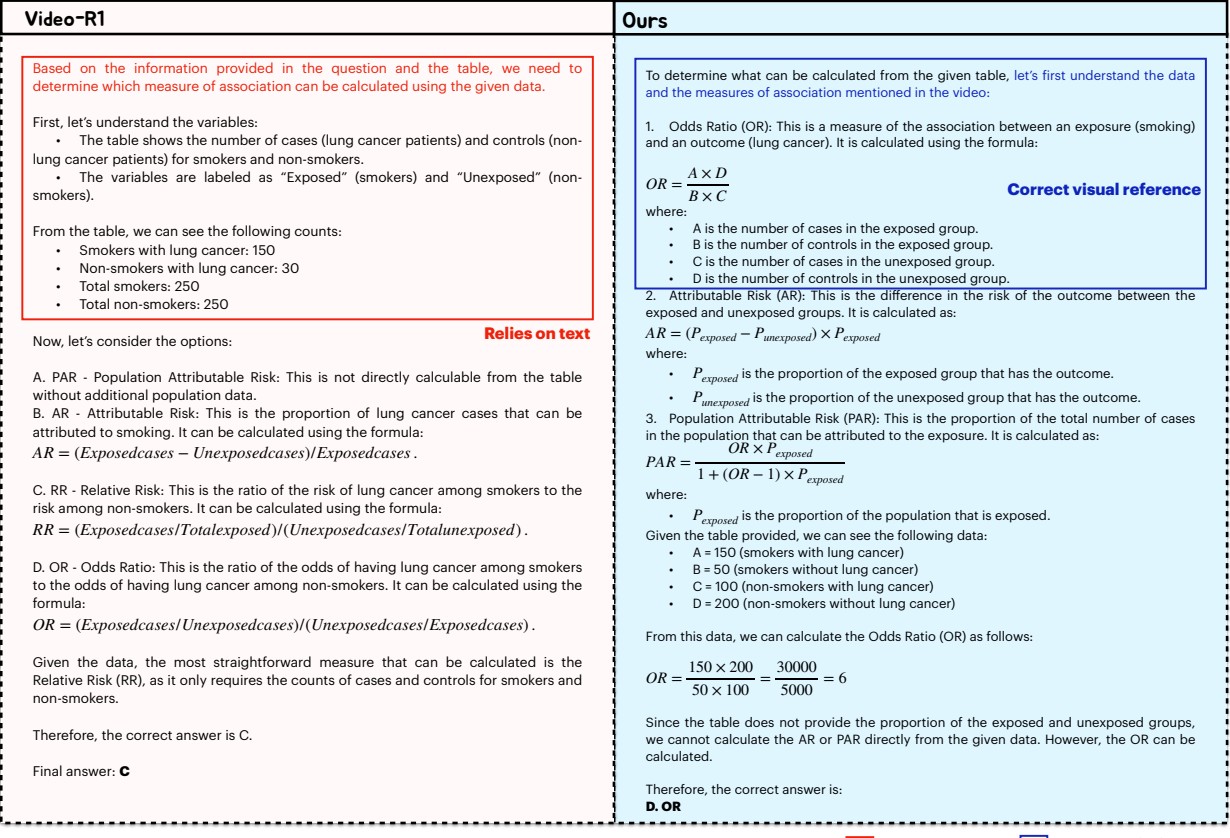

Figure 17: Public health question. Same comparison setup as Figure 13. This instance includes a reference image as part of the question prompt. VIDGROUND grounds its analysis in video-presented concepts of $OR$ (Odds Ratio), $PAR$ (Population Attributable Risk), and $AR$ (Attributable Risk), while Video-R1 (Feng et al., 2025) analyzes the question and options using prior knowledge.

**Video**

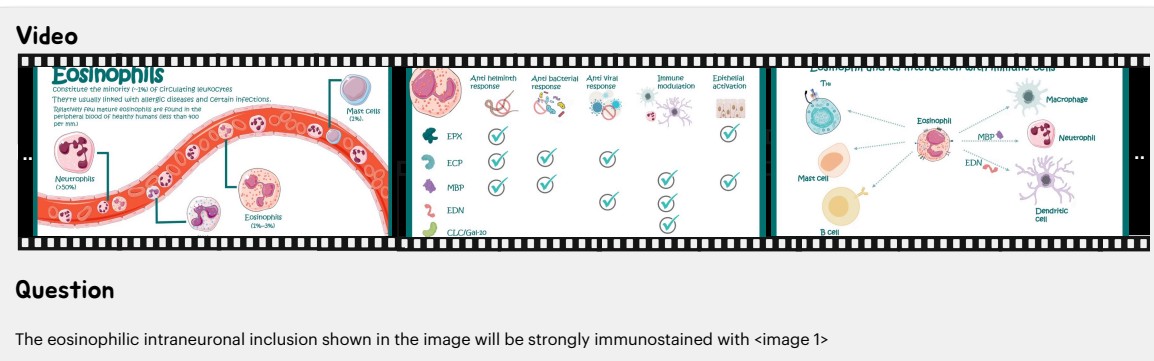

**Question**

The eosinophilic intraneuronal inclusion shown in the image will be strongly immunostained with <image 1>

A. tau
B. alpha synuclein
C. ubiquitin
**D. both B and C**

| Video-R1 | Ours |
|---|---|
| To determine which protein will strongly immunostain the eosinophilic intraneuronal inclusion, we need to consider the characteristics of the inclusion and the proteins listed.

1. tau: Tau is a microtubule-associated protein that is often associated with neurodegenerative diseases such as Alzheimer's disease. It is not typically found in eosinophilic intraneuronal inclusions.
2. alpha synuclein: Alpha synuclein is a protein involved in Parkinson's disease and other neurodegenerative disorders. It can form aggregates in neurons, similar to the eosinophilic inclusions seen in some neurological conditions.
3. ubiquitin: Ubiquitin is a small protein that is involved in the degradation of damaged proteins. It can also form aggregates in neurons, similar to the eosinophilic inclusions.
4. both B and C: This option suggests that both alpha synuclein and ubiquitin will strongly immunostain the inclusion. While both proteins can form aggregates in neurons, they are not typically associated with eosinophilic inclusions in the same way that alpha synuclein is.

Given the context of the image and the typical association of eosinophilic inclusions with alpha synuclein, the most likely protein to strongly immunostain the inclusion is alpha synuclein.

**Relies on text**

Therefore, the correct answer is:

B. alpha synuclein

However, since the question asks for the protein that will strongly immunostain the inclusion, and considering the typical association, the most accurate answer is:

D. both B and C

But since the question asks for the most specific and commonly associated protein, the answer is:

**B. alpha synuclein** | The image you are referring to is not provided in the text, but based on the context of the video content, which discusses the eosinophil and its role in immunity, the question is likely related to the staining of intraneuronal inclusions in eosinophils.

Eosinophils are known to contain granules that can be stained with certain markers. The options provided are:

A. tau
B. alpha synuclein
C. ubiquitin
D. both B and C

Given the context of the video, which focuses on eosinophils and their role in immunity, the most relevant staining markers for eosinophils are likely to be ubiquitin and alpha synuclein, as these are commonly used to stain intraneuronal inclusions in various diseases.

**Correct visual reference**

Therefore, the correct answer is:
**D. both B and C** |

▢ : Over-reliance on text   ▢ : Correct visual reference

Figure 18: Diagnostics and laboratory medicine question. Same comparison setup as Figure 13. This instance includes a reference image as part of the question prompt. VIDGROUND anchors its analysis in video content about eosinophils and their role in immunity, while Video-R1 (Feng et al., 2025) applies general knowledge without referencing the video.

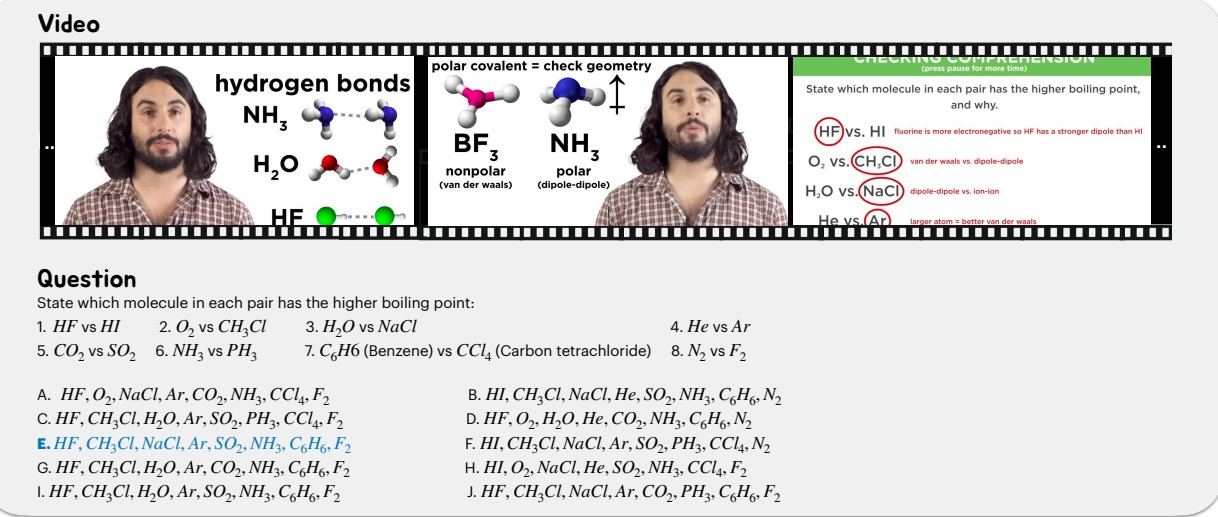

Figure 19: Chemistry question. Same comparison setup as Figure 13. VIDGROUND references video explanations of intermolecular forces and applies them to each molecular pair, while Video-R1 (Feng et al., 2025) directly analyzes the options using prior knowledge.

**Video**

**Question**

When drawing the influence line of any quantity value, assume that the load is ()<image 1>

**A. unit moving load with unchanged direction**
B. moving load
C. dynamic load
D. movable load

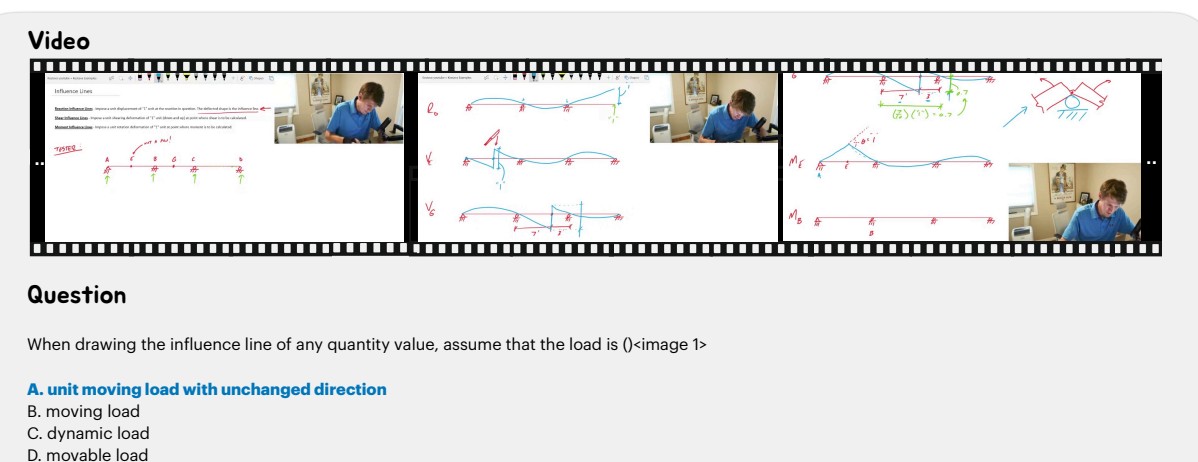

| Video-R1 | Ours |
|---|---|
| To determine the correct assumption when drawing the influence line of any quantity value, let's break down the options provided: **Relies on text**

A. unit moving load with unchanged direction - This option suggests that the load is a unit moving load with its direction remaining constant. This is a common assumption in structural analysis where the load is considered as a unit force that moves along a specific path.

B. moving load - This option is too broad. While it is true that the load is moving, the specific nature of the movement (unit or not) and the direction change need to be considered.

C. dynamic load - This option implies that the load has a time-dependent behavior, which is not necessarily the case when drawing an influence line. The influence line is typically used for static loads.

D. movable load - This option is similar to option B. It is too general and does not specify the nature of the movement or the direction change.

Given the context of structural analysis and the typical assumptions made when drawing influence lines, the most accurate and commonly used assumption is that the load is a unit moving load with unchanged direction.

Therefore, the correct choice is A.

Final answer: A

Let's verify the reasoning one more time. The influence line is often drawn under the assumption of a unit moving load with a fixed direction, which aligns with option A. This assumption simplifies the analysis and allows for the determination of the maximum values of the quantity of interest.

Final answer: **A** | To determine the correct answer, let's analyze the context of the video and the question.

The video is about influence lines in structural engineering, which are used to determine the maximum values of various quantities (like shear force, bending moment, etc.) under different loading conditions. The influence line is a graphical representation of the variation of a quantity with position of a unit load. **Correct visual reference**

The key point here is that the influence line is derived by considering a unit load moving along the structure. This unit load is typically assumed to be a unit moving load with an unchanged direction, as it helps in understanding the maximum values of the quantity of interest at different points along the structure.

Given this context, the correct answer is:

**A. unit moving load with unchanged direction**

This is because the influence line is constructed by considering a unit load moving along the structure, and the direction of the load is typically assumed to remain constant to simplify the analysis. |

☐ : Over-reliance on text   ☐ : Correct visual reference

Figure 20: Structural engineering question. Same comparison setup as Figure 13. This instance includes a reference image as part of the question prompt. Both models reach the correct answer (A. unit moving load with unchanged direction) through different reasoning paths. VIDGROUND references video content about influence lines in structural engineering and derives the answer from the demonstrated concept, while Video-R1 (Feng et al., 2025) eliminates options through prior knowledge without referencing the video. This example demonstrates that correct answers can emerge from either genuine video understanding or linguistic shortcuts, highlighting the importance of evaluating reasoning paths beyond accuracy metrics alone.

Table 8: Results on the Video-MME v2 leaderboard (no subtitles, visual frames only). *Frames* is the number of input frames per video; *Avg Acc* is the average accuracy over all QA pairs. Best per group in **bold**. Highlighted rows report our text-only probes of Gemini-3.1-Pro and GPT-5.4, where the model sees only the question text and no visual input; (+x) denotes the absolute gain over the Random Guess baseline (12.6%). Both probes score far above chance, showing that a substantial fraction of Video-MME v2 questions can be solved from language priors.

| Model | Frames | Avg Acc (%) |
|---|---|---|
| Random Guess | - | 12.6 |
| *Proprietary Models* | | |
| Gemini-3.1-Pro | 0 | 28.9 (+16.3) |
| GPT-5.4 | 0 | 23.6 (+11.0) |
| Gemini-3-Pro | 1 fps | **56.8** |
| Doubao-Seed-2.0-Pro-260215 | 64 | 53.1 |
| Gemini-3-Flash | 1 fps | 52.4 |
| MiMo-v2-Omni | 1 fps | 47.1 |
| Kimi-K2.5 | 64 | 46.0 |
| GPT-5 | 50 | 44.7 |
| *Open-Source Models* | | |
| Qwen3.5-397B-A17B | 64 | **35.0** |
| Qwen3-VL-235B-A22B | 64 | 33.8 |
| InternVL3_5-241B-A28B | 64 | 32.9 |
| KimiVL-16B-A3B | 64 | 32.3 |
| Qwen3.5-27B | 64 | 31.7 |
| Qwen2-5-VL-72B | 64 | 30.3 |
| Qwen3.5-122B-A10B | 64 | 29.5 |
| Qwen3.5-35B-A3B | 64 | 28.6 |
| Qwen3.5-9B | 64 | 26.0 |

