# OpenReview forum: "Watch Before You Answer: Learning from Visually Grounded Post-Training"
_TMLR — Under review for TMLR_

### Review · Reviewer_n216 · 2026-07-03

**Summary Of Contributions:**

This paper reveals that current progress in video understanding for Vision-Language Models (VLMs) is heavily inflated by "linguistic shortcutting," where 40-60% of questions in standard benchmarks and post-training datasets can be correctly answered using only text cues without ever watching the video. To address this pervasive language bias, the authors introduce VIDGROUND, a straightforward data curation strategy that filters post-training datasets to retain only genuinely visually grounded questions.

By training on this filtered dataset, which uses just 69.1% of the original data, with a standard reinforcement learning algorithm (GRPO), the authors demonstrate that models achieve consistent performance gains  across multiple benchmarks. Ultimately, the study highlights that curating high-quality, visually dependent data is a more effective way to advance true multimodal comprehension than many novel post-training algorithms.

**Audience:**

Yes

**Audience Explanation:**

yes, it demonstrates that data is crucial for the post-training of VLMs.

**Claims And Evidence:**

No

**Claims Explanation:**

The claims in this paper are partially supported.


> Supported claims:

1. The authors clearly demonstrate that the main performance gain of a stronger model comes from the text side.
2. They show concrete examples from four categories of TA questions, along with their statistical distribution in evaluation benchmarks.
3. Experiments show that a model trained on the VidGround dataset achieves better performance than using more novel post-training algorithms.


> Unsupported claims:

This paper claims, “However, despite many studies investigating this phenomenon for VQA, relatively little work has investigated it for video understanding.” Actually, many previous works have focused on the same research problem for video understanding, but were not discussed in this paper.


[1]: InstructionBench: An Instructional Video Understanding Benchmark, https://arxiv.org/pdf/2504.05040

[2]: A Shortcut-aware Video-QA Benchmark for Physical Understanding via Minimal Video Pairs, https://arxiv.org/pdf/2506.09987

**Requested Changes:**

The narrative of this work is interesting, specifically in how it compares the performance of data curation against other post-training algorithms. It successfully demonstrates that data is crucial for the post-training of VLMs. However, to improve the solidness of this paper, the authors should also carefully compare their approach with other data curation baselines. For example, although [1] focuses on image-text QA, its methods and baselines can be easily adapted to the video-text setting.

[1]: Focusing Where Vision Matters: Selective Training for Large Vision Language Models via Visual Information Gain, https://arxiv.org/abs/2602.17186

> Minor Suggestions:

In Table 2, many numbers are not shown properly.

---

### Review · Reviewer_emNF · 2026-07-04

**Summary Of Contributions:**

The authors perform an analysis of Video QA benchmarks, starting from solving them by using multimodal models without video input. They find that a significant amount of questions can be answered without watching the video. Multiple reasons are found, linguistic cues, common sense, by elimination of implausible answers, hallucinations (related to common sense).
As these questions do not require video processing, the authors hypothesize that they may hurt post training of VLMs, beyond muddying benchmark numbers. They thus propose a curation strategy to only post-train the models on visually grounded samples, improving the performance of Qwen2.5-VL-7B.

Main strengths:
- The authors perform an up-to-date analysis of text answerability in video QA benchmarks
- The authors turn this analysis into a curation pipeline for post-training
- The authors demonstrate improved performance over existing strategies

Main weaknesses:
- Empirical validation is a bit weak, one model and one dataset
- Claims need to be more carefully stated regarding questions being text-answerable. Notably, the use of already multimodal models adds - confounding factors to the analysis
- The text-answerability analysis is a bit cursory, considering only removing the video from the question (see below for suggestions)

**Audience:**

Yes

**Audience Explanation:**

The paper goes beyond previous more analytical works on text biases in benchmarks and turns this into a practical data curation method for post-training of video LLMs. This is a very interesting result for the community which can have significant practical implications.

While analyses of biases in multimodal models (see related works, and works pointed out below) are plenty, the authors perform it with more recent and state-of-the-art models, which also demonstrates that text-answerability is getting worse. Even benchmarks designed with text-answerability filtering using older models suffer from the problem.

**Broader Impact Concerns:**

I do not see particular concerns that should be addressed.

**Claims And Evidence:**

No

**Claims Explanation:**

My main concern is that the paper’s strongest claims conflate text-only correctness with true lack of visual dependence. The no-video setup convincingly shows that many questions are shortcutable by current models, but it does not convincingly show that the video is unnecessary, that models ignore the video when it is available, or that benchmark progress causally comes from language priors. A question may be answered correctly without video due to linguistic cues, commonsense, answer-choice artifacts, benchmark contamination, multimodal pretraining priors, or chance. Conversely, a model may answer correctly without video while still benefiting from video when video is provided. The paper would be stronger if it framed TA as an operational shortcutability measure and added counterfactual controls such as question-only, answer-only, wrong-video, irrelevant-video, single-frame, pure-LLM, and human no-video baselines.

The evidence for the benefits in post-training is in general more convincing, even if only one dataset and model are used. More evidence would still be beneficial to the paper to strengthen this aspect, which is the one that differs most from prior work.

**Requested Changes:**

In general, one of the two axes of the paper (analysis, improved post-training) should be improved to strengthen the paper. Precise comments and suggestions below.

**Critical**:

1) Along the work, models are used in text only modes to measure over-reliance on text. However, these models have already been trained on video and may have internalized some information through this process such as visual regularities, video priors, common video-question patterns, or even benchmark-related content. I can see in table 6 text-only models such as GPT-oss which reinforce the findings of the paper without this confounding factor. I believe that this use of multimodal models in text only mode should be discussed more, and text only models should be showcased more prominently to demonstrate the issues of the benchmark.

2) I think sometimes “does not use the video” and “can be solved without the video” are conflated. For example in Figure 2.a top-right, the climbing question can indeed be answered by common sense, but are we sure that the model does not use the video to answer it ? In general “rope” is a common sense answer, and probably the most likely one, but the use of video may make the model more confident in the prediction.

Related, in Figure 4 “since the question asks for the most prominent, and considering the variety of lines and shapes, the answer leans towards:” is flagged as an over-reliance on text. But this is a perfectly normal behaviour to look at the question, something that humans also do when the answer is ambiguous. Reasoning traces are not hard evidence that the model ignored the video.

In general, I wonder if the filtering may not be too conservative. Perhaps a human study could help determine when the question is really text answerable, and if the curation is consistent with what humans would determine as such.


**Nice to have**:
1) Currently, the post-training approach is only applied on Qwen2.5-VL. Ensuring its effectiveness on other base models would help strengthen the applicability of the method.

2) Missing related work on video QA benchmark biases. In [1] the authors perform similar analyses, using (weaker) text-only models, using the video and just the answers, or with a single frame. This is very related to the analyses done in the current paper, with deeper insights on various types of biases. Including similar, more diverse, bias analyses would strengthen the current work

3) The main VidGround filter removes a sample if GPT-5-mini answers correctly without the video. But in multiple-choice QA, some correct no-video answers will be random guesses. This means that some real visual questions may be removed by accident. The authors partially address this with multi-model and circular-evaluation variants, VidGround-M1 and VidGround-M2, but those are not the main method, and lead to poorer performance. Estimating the false-positive rate of the TA filter would be nice, perhaps by leveraging the multi-model filtering of -M1 and -M2



**Questions**:
1) What is the difference between “External knowledge” and  “Imagined (hallucinated) video content” text-answerable questions. It seems that they are both external knowledge but either with biases that can come only from text, or that can come from video ? would a text only model only flag External knowledge questions ?

2) In table 3, post-training on the full dataset consistently degrades performance. This is rather concerning and would suggest a poor quality dataset. One would expect a lesser increase, but not a degradation. Do the authors have an explanation for this ?



[1] Krojer, Benno, et al. "A shortcut-aware video-qa benchmark for physical understanding via minimal video pairs." TMLR (2025).

---

### Review · Reviewer_LFbo · 2026-07-12

**Summary Of Contributions:**

This paper investigates linguistic shortcutting in video understanding models. The authors show that several models achieve substantially above-chance accuracy on VideoMME, VideoMMMU, and MMVU when provided only with the questions and answer choices, without video input. Based on this observation, they propose VidGround, which uses GPT-5-mini to filter Video-R1-260K and retains only questions that cannot be answered in the text-only setting. The resulting data are then used to post-train Qwen2.5-VL-7B with GRPO. VidGround uses 69.1% of the original data but outperforms training on the full dataset, with gains of up to 6.2 percentage points in average full-benchmark accuracy.
The problem is important, and the proposed method is simple and practical. The experiments cover multiple models, benchmarks, and frame settings. However, the current TA/VG definition, experimental controls, and statistical analysis are not yet sufficient to fully support the main claims.

**Audience:**

No

**Audience Explanation:**

Linguistic shortcutting affects both the validity of video benchmarks and the quality of post-training data, making this an important problem for multimodal learning. The systematic text-only analysis across several models and benchmarks is informative, and the proposed filtering approach has practical potential. Even though the current TA/VG definition requires stronger validation, the findings should interest researchers working on video understanding, data-centric learning, and multimodal evaluation.

**Broader Impact Concerns:**

I do not identify major ethical concerns requiring escalation. However, a short discussion should cover biases and reproducibility limitations introduced by proprietary filtering models, the licensing and privacy of the source videos, and the risk of interpreting benchmark improvements as evidence of readiness for high-stakes applications such as medicine or autonomous driving.

**Claims And Evidence:**

No

**Claims Explanation:**

The paper presents promising evidence, but its central claims are not yet sufficiently supported.

First, a single correct or incorrect text-only prediction does not establish whether a question genuinely requires visual information. Random guessing, answer-position bias, prompt sensitivity, and possible training-data contamination can all affect the TA/VG labels. Agreement among several models is helpful, but correlated model failures may still reflect shared limitations rather than genuine visual necessity.
Second, filtering changes not only visual dependence but also dataset size, the image/video ratio, data sources, and difficulty. The paper lacks size-matched random subsets and controls matched by modality, source, and difficulty. Therefore, the observed gains cannot yet be attributed specifically to visual grounding.

Finally, improved accuracy, better frame scaling, and visual descriptions in generated reasoning do not directly demonstrate greater reliance on visual information. The paper also lacks multiple training seeds, confidence intervals, and significance tests. The headline 6.2-point improvement is measured against a Full-data model that substantially underperforms the base model; the smaller 2.2–2.4-point improvement over the base model should be reported equally clearly.

**Requested Changes:**

1. Validate the TA/VG labels.
Use repeated sampling and answer permutation for multiple-choice questions. Manually annotate a representative subset and report precision, recall, and inter-annotator agreement. Stable answerability should be distinguished from accidental correct or incorrect predictions.

2. Add matched controls.
Include a random subset with the same size as VidGround and subsets matched by image/video ratio, data source, task type, and difficulty. TA-only and different VG/TA mixture ratios would also be informative.

3. Directly measure visual reliance.
Compare full-video, text-only, randomly replaced video, shuffled-frame, and key-frame removal settings. Figure 1 should report question-level answer transitions rather than stacking text-only accuracy with the difference between two aggregate accuracies.

4. Report statistical uncertainty.
Run the main experiments with multiple random seeds and report standard deviations, confidence intervals, or significance tests.
Critical: Clarify the training setup.
Explain the relationship between “700 steps” and “one epoch,” and report the effective number of samples, global batch size, reward definitions, open-ended answer scoring, and exact model versions. It should also be clear whether asymmetric clipping is used in the main VidGround results.

5. Check for data overlap.
Analyze possible video and question overlap between Video-R1-260K and the three evaluation benchmarks, and repeat the evaluation after deduplication if necessary.
Recommended: Narrow the claims.
Until the labels are human-validated, VG should be described as “questions not solved by the specified text-only filter.” “Data quality is a major bottleneck” should be softened to “data quality may be an important bottleneck.”

---

### Review · Reviewer_jRe7 · 2026-07-16

**Summary Of Contributions:**

This paper asks whether recent video LLM progress reflects genuine visual understanding or is partly an artifact of language priors. The central observation is that a large fraction of questions in existing video QA benchmarks can be answered from text (question and options) alone, without watching the video. Building on this, the authors use a strong text-only model (GPT-5-mini) to filter out such "text-answerable" questions from a large training set (Video-R1-260K), retrain Qwen2.5-VL-7B on the filtered data, and argue that removing shortcut questions changes how reported gains should be interpreted. They also introduce a VG-Avg metric on the "visually grounded" subset and analyze frame-scaling on the full vs. VG splits.

Strengths: the core question is timely and relevant; the shortcut observation is intuitive and quantitatively supported across benchmarks; the filtering is simple and directly actionable for both training and evaluation.

Weaknesses: the Video-R1 baseline does not match the original paper (large, bidirectional gaps); the training recipe does not appear to include an SFT-CoT cold start, leaving unclear whether the effects come from filtering or a misaligned pipeline; VG-Avg is defined by the same model used for filtering (circularity); and all conclusions rest on a single model.

(Elaborated in Requested Changes and the evidence section below.)

**Audience:**

Yes

**Audience Explanation:**

The question of how much current video LLM progress reflects genuine visual understanding versus exploitation of language priors and dataset shortcuts is of broad interest to the multimodal and video-understanding community. The paper's central observation that a large fraction of existing video QA benchmarks can be answered from text alone, and that filtering these shortcut questions changes how we should interpret reported gains is timely and relevant to anyone building or evaluating video LLMs. Even where I have concerns about specific claims and reproduction (raised separately under the evidence criterion), the findings themselves would be of interest to a meaningful segment of TMLR's audience.

**Broader Impact Concerns:**

I have no significant broader impact or ethical concerns with this work. It is a methodological study of shortcut behavior in video QA benchmarks and a data-filtering procedure; it introduces no new models, data-collection practices, or applications that raise obvious ethical risks, and does not require a dedicated Broader Impact Statement in my view.

**Claims And Evidence:**

No

**Claims Explanation:**

Not in the current version. My concerns are concentrated on the central empirical claims:

1. Reproduction mismatch. The reported Video-R1 baseline does not match the original paper, in both directions and by large margins (e.g. at 64 frames, MMVU 53.2 vs. the original 63.8; Video-MMMU 52.4 vs. the original 45.4). Since the paper's argument rests on this comparison, these discrepancies must be reconciled before the conclusions can be trusted.

2. Missing SFT-CoT cold start. I could not find any mention of the CoT cold-start stage, which is a key step before RL in the original Video-R1 recipe. If it was skipped, this alone could explain both the misaligned numbers and the observed performance drop from full-data training, undermining the central claim.

3. "Cannot be answered from text" ≠ "needs vision." The filtering treats these as equivalent, but they are not; the resulting "visually grounded" subset likely contains noise, and this equivalence is asserted rather than justified.

4. Non-independent evaluation. The same GPT-5-mini model is used both to filter the data and to define the VG-Avg test subset, which risks a circular setup where gains may reflect alignment with the filter rather than genuine visual grounding.

5. Single-model validation. All conclusions rest on one model (Qwen2.5-VL-7B); it is unclear whether they generalize to other families or scales.

I would be willing to raise my assessment if the authors (a) reconcile the Video-R1 reproduction and confirm the full training pipeline including the CoT stage, (b) justify or qualify the text-answerability / vision-requirement equivalence, and (c) provide an evaluation subset that is independent of the filtering model.

**Requested Changes:**

I group the requested changes below into [Critical] (must be addressed to secure my recommendation for acceptance) and [Strengthening] (would improve the work but are not decisive).

[Critical] 1. Reconcile the Video-R1 reproduction.
The reported Video-R1 baseline does not match the original paper, in both directions and by large margins. For example, at 64 frames, MMVU is reported as 53.2 here vs. 63.8 in the original, and Video-MMMU as 52.4 here vs. 45.4 in the original. Because the paper's central argument rests on comparing against this baseline, please report the full evaluation protocol used for Video-R1 (number of frames, prompt template, decoding settings, and exact checkpoint) and reconcile these discrepancies. This is critical: if the baseline is not faithfully reproduced, the conclusions drawn from the comparison cannot be trusted.

[Critical] 2. Clarify the training recipe, in particular the SFT-CoT cold start.
I could not find any mention of the SFT chain-of-thought (CoT) cold-start stage, which is a key step before RL in the original Video-R1 recipe. Please state explicitly whether this stage was included. If it was omitted, this alone could plausibly explain both the misaligned baseline numbers and the observed performance drop from full-data training, which would undermine the central claim. Please confirm that the training pipeline matches the original recipe, or clearly characterize where and why it differs.

[Critical] 3. Justify or qualify the equivalence between "cannot be answered from text" and "requires vision."
The filtering treats a question that a text-only model cannot answer as one that requires visual understanding. To be precise, these are not the same. A question may be unanswerable from text because it is ambiguous, under-specified, requires external or common-sense knowledge, or the text-only model simply happens to fail on it, without genuinely requiring vision. Please either justify this equivalence or explicitly frame it as an approximation and characterize the resulting noise in the "visually grounded" subset (e.g. via human verification of a sample).

[Critical] 4. Address the non-independence of the VG-Avg evaluation.
The same GPT-5-mini model is used both to filter the training data and to define the "visually grounded" (VG-Avg) evaluation subset. This risks a circular setup in which improvements on VG-Avg partly reflect alignment with the filter rather than genuine visual grounding. Please provide an evaluation subset whose definition is independent of the filtering model, for example defined by a different model, or by human annotation, and report results on it.

[Strengthening] 5. Validate on more than one model.
All the main conclusions rest on a single model (Qwen2.5-VL-7B). Different architectures, scales (e.g. 32B/72B), and training recipes may exhibit different shortcut behavior and different sensitivity to the filtering. Adding at least one additional model family or scale would substantially strengthen the generality of the claims. I consider this strengthening rather than critical, but the paper's contribution would be much more convincing with it.

[Strengthening] 6. Discuss how the method scales as the base model approaches the filter's capability.
The approach implicitly requires the text-only filter to be stronger than the model being evaluated, so that it can reliably identify text-answerable questions. As target models approach or exceed the filter's capability, this boundary becomes unreliable. A discussion (and ideally an experiment) of how the filtering behaves in this regime would clarify the method's applicability to frontier models.

[Strengthening] 7. Report evaluation robustness.
Where feasible, please report variability (e.g. confidence intervals or multiple seeds) for the main comparisons, so that readers can judge whether the reported gaps are statistically meaningful rather than within run-to-run noise.